# InfoCon: Concept Discovery with Generative and Discriminative Informativeness

**Ruizhe Liu**[1,2*]**, Qian Luo**[1]**, Yanchao Yang**[1,3]
[1]HKU Musketeers Foundation Institute of Data Science, The University of Hong Kong
[2]School of Electronics Engineering and Computer Science, Peking University
[3]Department of Electrical and Electronic Engineering, The University of Hong Kong
lrz360@stu.pku.edu.cn, {yanchaoy,qianluo}@hku.hk

## Abstract

We focus on the self-supervised discovery of manipulation concepts that can be adapted and reassembled to address various robotic tasks. We propose that the decision to conceptualize a physical procedure should not depend on how we name it (semantics) but rather on the *significance* of the informativeness in its representation regarding the low-level physical state and state changes. We model manipulation concepts – discrete symbols – as generative and discriminative goals and derive metrics that can autonomously link them to meaningful sub-trajectories from noisy, unlabeled demonstrations. Specifically, we employ a trainable codebook containing encodings (concepts) capable of synthesizing the end-state of a sub-trajectory given the current state – *generative informativeness*. Moreover, the encoding corresponding to a particular sub-trajectory should differentiate the state within and outside it and confidently predict the subsequent action based on the gradient of its discriminative score – *discriminative informativeness*. These metrics, which do not rely on human annotation, can be seamlessly integrated into a VQ-VAE framework, enabling the partitioning of demonstrations into semantically consistent sub-trajectories, fulfilling the purpose of discovering manipulation concepts and the corresponding sub-goal (key) states. We evaluate the effectiveness of the learned concepts by training policies that utilize them as guidance, demonstrating superior performance compared to other baselines. Additionally, our discovered manipulation concepts compare favorably to human-annotated ones while saving much manual effort. Our code is available at: https://zrllrz.github.io/InfoCon_/

## 1 Introduction

Conceptual development is of core importance to the emergence of human-like intelligence, ranging from primary perceptual grouping to sophisticated scientific terminologies (Sloutsky, 2010). We focus on embodied tasks that require interaction with the physical environment and seek the *manipulation concepts* that are critical for learning efficient and generalizable action policies (Lázaro-Gredilla et al., 2019; Shao et al., 2021). Recently, Large Language Models (LLMs) have enabled many interesting robotic applications with their reasoning capabilities that can break complex embodied tasks into short-horizon interactions or manipulation concepts (Singh et al., 2023). However, LLMs are trained with an internet-scale corpus, representing a vast amount of linguistic knowledge of manipulations, but *lack* embodied experiences that *ground* these manipulation concepts to specific physical states (Brohan et al., 2023; Huang et al., 2023). In contrast, it is worth noting that in human development, infants initially acquire physical skills, e.g., crawling, grasping, and walking, before delving into the complexities of language, which grant the manipulation concepts described in language with groundingness in the first place (Walle & Campos, 2014; Libertus & Violi, 2016).

Drawing inspiration from the human developmental process, we aim to equip embodied agents with manipulation concepts *intrinsically connected* to physical states without relying on additional grounding techniques. *Specifically*, we seek algorithms that allow agents to abstract or identify

---

*Work done as a research assistant at HKU.

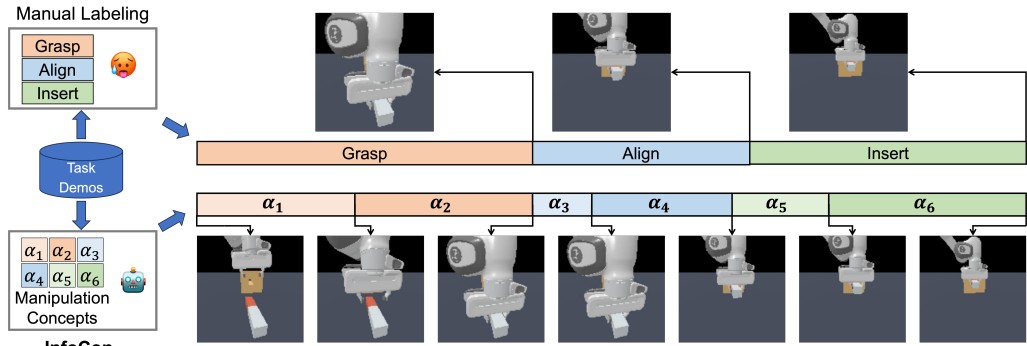

Figure 1: The proposed generative and discriminative informativeness and the derived *InfoCon* algorithm can discover manipulation concepts from noisy, unlabeled demonstrations. Each identified concept relates to a sub-goal and defines the partitioning of a whole trajectory into sub-trajectories, showing the process governed by the concept for achieving the sub-goal. Concepts from InfoCon share similarities with human-annotated ones while having more fine-grained semantics, which can be more beneficial for physical interactions but are time-consuming to label manually.

manipulation concepts from their embodied experiences, such as demonstration trajectories, without humans specifying and annotating the involved concepts. The discovery process serves a dual purpose. *Firstly*, it generates a set of discrete symbols that hold semantic meaning, with some potentially representing concepts like "grasping a block" or "aligning a block with a hole on the wall." *Furthermore*, the discovery process should explicitly establish correspondences between the concepts and physical states, thereby achieving the grounding for low-level actions while minimizing annotation efforts.

Therefore, we propose to model manipulation concepts as *generative and discriminative goals*. Specifically, as a generative goal, a manipulation concept should help predict the goal state even though it has not been achieved yet. For example, when given the manipulation concept "grasp the block," one can already (roughly) synthesize how the scene looks when the robotic gripper grasps the block. We formulate it as the informativeness between the manipulation concept and the synthesized ending state when the interaction implied by the manipulation concept is accomplished, which we name *generative informativeness*. On the other hand, given a manipulation concept as a discriminative goal, one could tell if the current state is within the process of achieving the goal state implied by the concept. For example, with the concept "place the cup under the faucet," one would assign low compatibility to the state of "pouring water to the mug" compared to the correct one. Similarly, we can formulate it as the informativeness between the manipulation concept and the binary variable, indicating whether the state falls within the manipulation process, which we call *discriminative informativeness*. Moreover, as a discriminative goal, a manipulation concept should inform the following action. If an action incurs a higher discriminative score (compatibility), then it should be executed to accomplish the task. Accordingly, we formulate it as the informativeness between the *gradient* of the discriminative function (conditioned on the manipulation concept) and the next action in the demonstration.

With the proposed metrics, we can train a VQ-VAE (Van Den Oord et al., 2017) architecture to learn the discrete representations of potential manipulation concepts from the demonstration, as well as the assignment (grounding) between the learned concepts and the physical states, even though *no concept descriptions are available* in any form. We examine the quality of the learned manipulation concepts through the automatically acquired grounding, and verify that these concepts do come with semantic meaning in terms of human linguistics Fig 1. We further demonstrate the usefulness of the learned manipulation concepts by using the grounded states as guidance to train manipulation policies. Experimental results show that the discovered manipulation concepts from unannotated demonstrations enable policies that surpass the state-of-the-art counterparts, while achieving comparable performance with respect to the oracle trained with human labels, which verifies the effectiveness of the proposed metrics and training for *self-supervised manipulation concept discovery*.

## 2 METHOD

**Problem Setup**   We aim to characterize the multi-step nature of *low-level* manipulation tasks. More explicitly, we assume access to a set of pre-collected demonstrations or trajectories, i.e.,

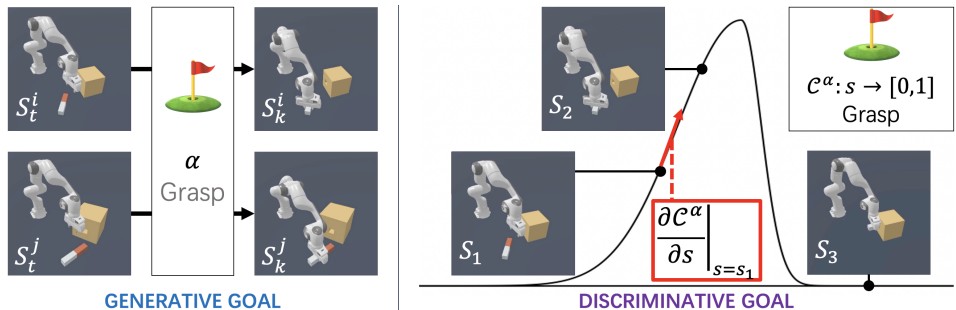

Figure 2: We characterize a manipulation concept from two conjugate perspectives. As a generative goal, a manipulation concept helps synthesize the state when the physical process meant by the concept is accomplished. On the other, as a discriminative goal, a manipulation concept indicates (through a scoring function) whether a state lies within the process governed by it. Moreover, it informs the next action with the gradient of the scoring function, as the action taken should maximize the discriminative utility.

$\mathcal{D} = \{\tau_i\}_{i=1}^{N}$ and $\tau_i = \{(s_t^i, a_t^i)\}_{t=1}^{T(i)}$, which is a sequence of state-action pairs. Our *goal* is to partition each trajectory into semantically meaningful segments (continuous in time), while maintaining consistency between trajectories of the same task without resorting to human annotations. We call what *governs the actions* within a particular trajectory segment the *manipulation concept*. And the "semantic meaning" of the manipulation concept lies in its generative and discriminative informativeness, which we discuss in Sec. 2.1. We further name the state at the boundary of two segments as the key state (Jia et al., 2023). We examine how the learned manipulation concepts align with human semantics, as well as the quality of the partitions, by evaluating the effectiveness of the derived key states on physical manipulation benchmarks. We adopt the CoTPC Jia et al. (2023) framework to train concept-guided policies across different benchmarks. Next, we elaborate on the proposed partitioning metrics.

## 2.1 MANIPULATION CONCEPT AS GENERATIVE AND DISCRIMINATIVE GOALS

Suppose a task can be described by $K$ manipulation concepts $\{\alpha_k\}_{k=1}^{K}$ with $\alpha_k \in \mathbb{R}^M$. Or equivalently, a trajectory $\tau = \{(s_t, a_t)\}_{t=1}^{T}$ from this task could be divided into $K$ segments $\{\beta_k\}_{k=1}^{K}$, with each $\beta$ a continuous short-horizon trajectory and $\tau = \cup_{k=1}^{K} \beta_k$. Note that we do not endow $\alpha_k$ with any specific linguistic descriptions at this moment (e.g., "pick up a cup" or "place the cup under the faucet"). Instead, we treat them as symbols grounded in the segments with consistency across different trajectories of the same task. In other words, we assume the existence of a partitioning function (neural network) $\Phi$ such that $\{\beta_k\}_{k=1}^{K} = \Phi(\tau)$. We detail the structure of $\Phi$ later and enable the training of $\Phi$ by proposing the following criteria.

**As a generative goal,** a manipulation concept $\alpha_k$ shall, given the current state, inform the end state of $\beta_k$, i.e., the key state $s_{t^k}$ with $t^k = \sum_{j=1}^{k} |\beta_j|$, where $|\cdot|$ is the length of a (sub)sequence. For example, with the manipulation concept that resembles "grasp the mug handle," one can imagine the picture depicting when the handle is firmly grasped. We formalize this metric using Shannon's mutual information:

$$\mathcal{L}^{\text{gen}}(\boldsymbol{\alpha}) = \mathbb{I}(\boldsymbol{\alpha}; \boldsymbol{s}^{\text{key}}|\boldsymbol{s}), \boldsymbol{s}^{\text{key}} \in \cup_{\tau_i}\{s_{t^k}(\tau_i)\}_k. \quad (1)$$

We name the above the **generative informativeness** since knowing the manipulation concept can help confidently synthesize the imminent key state after $\boldsymbol{s}$. Please note that the manipulation concept, key state, and state are random variables depending on the trajectory, which is omitted for simplicity.

**As a discriminative goal,** a manipulation concept $\alpha_k$ should tell whether the current state is within the process described by $\alpha_k$ or not. For example, in the task to get water to drink, if $\alpha_k$ resembles "approach the faucet," then the state before a cup is grasped or the state "drinking water" should have low compatibility with $\alpha_k$ in contrast to the states within the corresponding sub-trajectory of approaching the faucet. Thus, the knowledge of the manipulation concept $\alpha_k$ helps distinguish the states governed by it from the other states. We characterize this phenomenon by instantiating a **compatibility function**:

$$\mathcal{C} : \boldsymbol{\alpha} \times \boldsymbol{s} \to [0, 1], \quad (2)$$

such that values close to 1 imply high compatibility of the state $s$ with the process described by the manipulation concept $\alpha$. We can also write $\mathcal{C}^{\alpha}(\cdot)$ to reflect the fact that the manipulation concept

indexes a discrimination function, hence the role of $\alpha$ as a discriminative goal and we call Eq. 2 the **discriminative informativeness** of the manipulation concept.

Furthermore, we propose that the gradient of the compatibility function $\mathcal{C}^\alpha$ should be informative of the next action $a$. We believe, as a densely defined function, it is beneficial that $\mathcal{C}^\alpha$ not only indicates "what" the agent is doing but also "how" to do it by modulating the fluctuations of the compatibility function around $s$. Still, considering the concept of "filling the cup with water," the action of putting the cup under the faucet shall be assigned higher compatibility than the action of moving the cup toward a microwave. This illustrates that the compatibility function shall inform state changes (induced by actions) via the changes in its value, depicted through its gradient. Accordingly, we formulate this characteristic of $\mathcal{C}^\alpha$ as:

$$\mathcal{L}^{\mathrm{dis}}(\boldsymbol{\nabla}\mathcal{C}) = \mathbb{I}(\boldsymbol{\nabla}\mathcal{C}; \boldsymbol{a}|\boldsymbol{\alpha}), \tag{3}$$

where $\nabla\mathcal{C} = \dfrac{\partial \mathcal{C}^\alpha}{\partial s}$, given the manipulation concept $\alpha$. Consequently, we name Eq. 3 the **actionable informativeness** of the manipulation concept as a discriminative goal. We illustrate the idea of a manipulation concept as both generative and discriminative goals in Fig. 2, which shows different aspects during a manipulation process as represented by the proposed informativeness criteria.

*In summary,* we propose that a manipulation concept $\alpha$ should 1). inform the imminent key (physical) state given the current state as a generative goal; 2). indicate whether the current state is within the process governed by itself as a discriminative goal, as well as 3). inform the action with the gradient of the instantiated discriminative function $\mathcal{C}^\alpha$. Please note that these metrics do not count on human supervision. In other words, as long as we have a network $\Phi$ that takes in a manipulation trajectory and outputs a set of continuous segments, we can train $\Phi$ using the proposed metrics to discover manipulation concepts (consistent partitions) shared across trajectories of different tasks. Next, we operationalize the self-supervised manipulation concept discovery by specifying the training architecture and the objectives.

## 2.2 Self-Supervised Manipulation Concept Discovery

**Network Structure of $\Phi$**    We adapt the basic structure proposed in VQ-VAE (Van Den Oord et al., 2017) to accommodate the sequential nature of a state-action sequence and the need for continuous sub-trajectories. Specifically, we employ a transformer encoder $\phi$ that maps state $s_t$ to a latent $z_t$ in an autoregressive manner:

$$z_t = \phi(s_t|\{(s_j, a_j)\}_{j=1}^{t-1}). \tag{4}$$

This autoregressive design alleviates the ambiguity in predicting the latent by supplying rich history information. Moreover, it helps smooth out noise in $z_t$ to facilitate the partitioning of the trajectory into continuous sub-trajectories. To further enhance the segmentation continuity, we devise a positional encoding scheme for time $t$ and apply it to the latent $z_t$. The proposed positional encoding can capture the fact that nearby latents have a good chance of being assigned to the same manipulation concept without causing over-smoothing. Please refer to Sec. A.1 for more details. In the following, we abuse $z_t$ for the latent appended with the proposed positional encoding.

**Trajectory Partitioning with Manipulation Concepts**    To further process $\{z_t\}_{t=1}^T$ and derive the partitioning, we instantiate $K$ trainable vectors, which serve as the manipulation concepts $\{\alpha_k\}_{k=1}^K$. We then assign the state or latent $z_t$ to the concept that shares the largest similarity. More explicitly, denote $\eta$ as the concept assignment function, then we have:

$$\eta(z_t) = \arg\max_k \mathrm{p}(z_t \rightarrow \alpha_k) = \arg\max_k \frac{\exp(\langle z_t, \alpha_k \rangle / \tau)}{\sum_{k=1}^K \exp(\langle z_t, \alpha_k \rangle / \tau)}, \tag{5}$$

where $\mathrm{p}(z_t \rightarrow \alpha_k)$ is the probability of assigning latent $z_t$ to the concept $\alpha_k$, and $\langle \cdot, \cdot \rangle$ is the cosine similarity between two vectors. This assignment process allows us to group the states into segments $\{\beta_k\}_{k=1}^K$, i.e., binding states corresponding to the same concept, and serves as a ground to elaborate the training objectives. Note that we also need to ensure the gradient flow during training, thus, we use a technique similar to the gradient preserving proposed in VQ-VAE (Van Den Oord et al., 2017). Please see Sec. A.5 for more details.

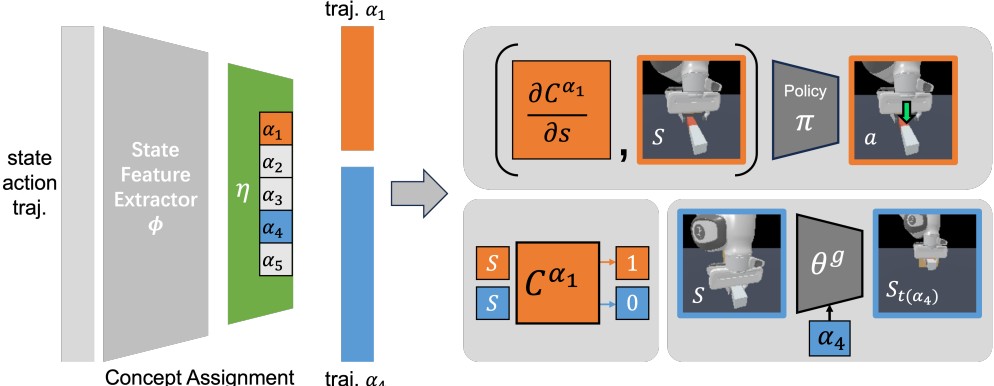

Figure 3: Training pipeline of InfoCon. Features extracted from the state-action trajectory (using $\phi$) are compared with learnable concepts, which are grounded according to Eq. 5. The generative informative loss (Eq. 6) trains $\theta^{\mathrm{g}}$ to predict the key (end) state of a sub-trajectory. The discriminative informative loss trains a compatibility function $\mathcal{C}$ conditioned on the concept (Eq. 7), which tells whether a state is compatible with the concept. Moreover, the actionable informativeness loss trains a policy $\pi$ for action prediction (Eq. 8). Together, these components enforce the grounding to be physically and semantically meaningful.

**Training Objectives**   With the derived sub-trajectories, we first locate the key state for every single state. Let $\alpha_t^i$ be the predicted concept – for trajectory $i$ at time step $t$ – from $\{\alpha_k\}_{k=1}^K$, and let $s_{t(\alpha_t^i)}^i$ be the (to be achieved) key state of $s_t^i$ from the trajectory $\tau_i = \{(s_t^i, a_t^i)\}_{t=1}^{T(i)}$. Then $s_{t(\alpha_t^i)}^i$ is determined by setting $t(\alpha_t^i) = \arg\min_u \{u \geq t, \alpha_u^i \neq \alpha_{u+1}^i\}$. Next, we detail the loss terms.

*Generative Goal Loss.* To maximize the generative informativeness (Eq. 1), we employ a reconstruction loss that is related to maximizing the mutual information (Hjelm et al., 2018). We instantiate a network $\theta^{\mathrm{g}}$ that predicts the key state from the current conditioned on the associated manipulation concept $\alpha_t^i$:

$$\mathcal{L}^{\mathrm{gen}}(\boldsymbol{\alpha}, \phi; \theta^{\mathrm{g}}) = \frac{1}{N} \sum_{i=1}^N \frac{1}{T(i)} \sum_{t=1}^{T(i)} \|\theta^{\mathrm{g}}(s_t^i; \alpha_t^i | \{(s_u^i, a_u^i, \alpha_u^i)\}_{u=1}^{t-1}) - s_{t(\alpha_t^i)}^i\|_2^2, \tag{6}$$

with $\|\cdot\|_2$ the $L$-2 norm. This term encourages $\alpha_t^i$ to be informative of the to-be-achieved key state by minimizing the prediction error.

*Discriminative Goal Loss.* According to the defining term (Eq. 2) of a manipulation concept as a discriminative goal, we instantiate a network $\mathcal{C}^{(\cdot)}$ as a hyper-classifier, which can decode manipulation concepts from $\{\alpha_k\}_{k=1}^K$ into compatibility functions representing the discriminative goal of these concepts. More specifically, $\mathcal{C}^{\alpha_t^i}$ should assign high scores to states in a trajectory $\tau^i$ that are governed by $\alpha_t^i$. We formulate it as a binary classification and employ the cross entropy loss to maximize the *discriminative informativeness* (Eq. 2):

$$\mathcal{L}_c^{\mathrm{dis}}(\boldsymbol{\alpha}, \phi; \mathcal{C}) = -\frac{1}{K} \sum_{k=1}^K \langle \frac{1}{|\{\alpha_t^i = \alpha_k\}|} \sum_{\alpha_t^i = \alpha_k} \log \mathcal{C}^{\alpha_k}(s_t^i) + \frac{1}{|\{\alpha_t^i \neq \alpha_k\}|} \sum_{\alpha_t^i \neq \alpha_k} \log(1 - \mathcal{C}^{\alpha_k}(s_t^i)) \rangle. \tag{7}$$

Note that the quantity inside $\langle \cdot \rangle$ is the classification error for a manipulation concept's classifier but is normalized to account for the imbalance between the positive and negative samples. Please see Sec. A.7 for details of the hyper-classifier $\mathcal{C}$.

To derive the loss for maximizing the *actionable informativeness* (Eq. 3), we instantiate a policy network $\pi$, which minimizes the prediction error of the action conditioned on the manipulation concept:

$$\mathcal{L}_a^{\mathrm{dis}}(\boldsymbol{\alpha}, \phi; \mathcal{C}, \pi) = \frac{1}{N} \sum_{i=1}^N \frac{1}{T(i)} \sum_{t=1}^{T(i)} \|\pi(s_t^i, \frac{\partial \mathcal{C}^{\alpha_t^i}}{\partial s}\Big|_{s=s_t^i}; h_t^i) - a_t^i\|_2^2, \tag{8}$$

where $h_t^i = \{(s_u^i, a_u^i, \frac{\partial \mathcal{C}^{\alpha_u^i}}{\partial s}\Big|_{s=s_u^i})\}_{u=1}^{t-1}$ is the history, and we use $L$-2 norm for the discrepancy. Please note that the policy employed here is mainly for the purpose of manipulation concept dis-

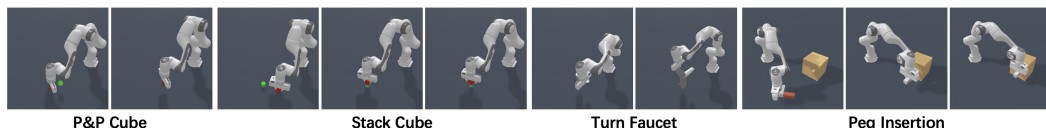

**P&P Cube**  **Stack Cube**  **Turn Faucet**  **Peg Insertion**

Figure 4: Examples of the manually defined key states (concepts) in different manipulation tasks. From left to right: **P&P Cube** and its two key states ("Grasp", "End"). **Stack Cube** and its three key states ("Grasp $A$", "$A$ on $B$", "End"). **Turn Faucet** and its two key states ("Contacted", "End"). **Peg Insertion** and its three key states ("Grasp", "Align", "End").

covery, which is different from the policy in downstream manipulation tasks (Please see Sec. 3.1 for more details). By minimizing the above quantity, the information in the partial derivative about the next action should be maximized. Further, following Van Den Oord et al. (2017), we add a term that encourages more confident assignments by Eq. 5, i.e.,

$$\mathcal{L}^{\mathrm{ent}}(\boldsymbol{\alpha}, \phi) = -\frac{1}{K}\sum_{k=1}^{K}\frac{1}{|\{\alpha_t^i = \alpha_k\}|}\sum_{\alpha_t^i = \alpha_k}\log(\mathrm{p}(z_t^i \nrightarrow \alpha_k)), \tag{9}$$

which is similar to minimizing the entropy of the assignment probability function $\mathrm{p}(z_t \nrightarrow \alpha_k)$.

Finally, the **total training loss** for self-supervised manipulation concept discovery with the proposed generative and discriminative informativeness metrics can be summarized as:

$$\mathcal{L} = \mathcal{L}^{\mathrm{gen}}(\boldsymbol{\alpha}, \phi; \theta^{\mathrm{g}}) + \mathcal{L}_a^{\mathrm{dis}}(\boldsymbol{\alpha}, \phi; \mathcal{C}, \pi) + \lambda(\mathcal{L}_c^{\mathrm{dis}}(\boldsymbol{\alpha}, \phi; \mathcal{C}) + \mathcal{L}^{\mathrm{ent}}(\boldsymbol{\alpha}, \phi)), \tag{10}$$

with $\lambda$ balancing the importance between the prediction and the classification terms. More implementation details can be found in Sec. A and Sec. B, and we study the effectiveness of each component in the experiments.

## 3 EXPERIMENTS

### 3.1 EXPERIMENTAL SETTINGS

We evaluate the effectiveness of InfoCon and the derived key states on four robot manipulation tasks from ManiSkill2 (Gu et al., 2023), an extended version of ManiSkill (Mu et al., 2021): **P&P Cube**: To pick up a cube, move it to a specific location, and keep it still for a short while. **Stack Cube**: To pick up a cube and put it on the top of another cube. **Turn Faucet**: To turn different faucets for enough angle values. **Peg Insertion**: To pick up a cuboid-shaped peg and insert it into a hole in a given object. We visualize the manipulation tasks with the manually defined and grounded key states from CoTPC (Jia et al., 2023) in Fig. 4.

**Training details.** Following the setup of CoTPC, we choose the same sets of trajectories (without ground truth key states) for the four tasks above for training and evaluation. Specifically, we collect 500 training trajectories for each task, 100 evaluation trajectories for P&P Cube and Stack Cube, 100 evaluation trajectories for seen faucets, 400 trajectories for unseen faucets, and 400 evaluation trajectories for Peg Insertion. With the pipeline shown in Fig. 3, we train the model and use the state encoder (Eq. 4) along with the concept assignment module (Eq. 5) to perform the partitioning and grounding of all trajectories. We choose the last state (of a sub-trajectory) as the key state of the corresponding manipulation concept. After collecting the discovered key states, we train a CoTPC policy optimizing the action prediction with the key states as guidance. We leave out the key state prediction in the policy training when the key state does not appear in the trajectory (e.g., concepts discovered for one task may not appear in another). Please see Sec. A and Sec. B for more details.

**Evaluation metrics.** The efficacy of our approach is mainly assessed by the task success rate of policies that are trained utilizing the key states identified via InfoCon. Additionally, for curiosity, we introduce a metric for a rough understanding of whether there are similarities between the predicted key states and the ground truth key states (designed by humans). Suppose the time steps of ground truth key states are $\{t_k^{gt}\}_{k=1}^{K'}$, and the time steps of predicted key states are $\{t_j\}_{j=1}^{K}$ (here $K'$ is not always equal to $K$). We propose the **Human Intuition Similarity**, or **HIS**, as in the following:

$$\mathbf{HIS}(\{t_j\}_{j=1}^{K}, \{t_k^{gt}\}_{k=1}^{K'}) = \sum_{k=1}^{K'}(t_{\arg\min_j (t_j \geq t_k^{gt})} - t_k^{gt}) \tag{11}$$

Table 1: Success rate (%) of different methods. For the task Turn-Faucet, we provide results on three kinds of situations: seen environments (s.&sf.), unseen environments with seen faucet types (u.&sf.), and unseen environments with unseen faucet types (u.&uf.)[1]. $\star$ marks key state labeling methods dependent on human semantics (zero-shot), and $\bullet$ marks key state labeling methods intrinsically related to the inherent properties of the data.

| SR(%) | P&P Cube | | Stack Cube | | Turn Faucet | | | Peg Insertion | |
|---|---|---|---|---|---|---|---|---|---|
| | seen | unseen | seen | unseen | s.&sf. | u.&sf. | u.&uf. | seen | unseen |
| Decision Transformer | 65.4 | 50.0 | 13.0 | 7.0 | 39.4 | 32.0 | 9.0 | 5.6 | 2.0 |
| Last State$^\star$ | 70.8 | 60.0 | 12.0 | 4.0 | 47.8 | 46.0 | 21.0 | 9.6 | 5.3 |
| AWE$^\bullet$ | 96.2 | 78.0 | 45.4 | 18.0 | 53.4 | 50.0 | 14.5 | 31.6 | 4.3 |
| LLM+CLIP$^\star$ | 92.2 | 71.0 | 44.6 | 21.0 | 35.8 | 31.0 | 18.0 | 63.2 | 13.5 |
| GT Key States$^\star$ | 75.2 | 70.0 | 58.8 | 46.0 | **56.4** | **57.0** | **31.0** | 52.8 | 16.8 |
| **InfoCon**$^\bullet$ | **96.6** | **81.0** | **63.0** | **47.0** | 53.8 | 52.0 | 17.8 | **63.6** | **17.8** |

Table 2: Human Intuition Score (**HIS**) of different discovery/grounding methods.

| HIS | P&P Cube | Stack Cube | Turn Faucet | Peg Insertion |
|---|---|---|---|---|
| LLM+CLIP | 57.51 | 93.19 | 84.39 | 164.29 |
| AWE | 0.003 | 2.49 | 14.72 | 9.25 |
| InfoCon | 2.60 | 8.98 | 18.03 | 9.05 |

The **HIS** metric quantifies the temporal distance between a given ground truth key state and the nearest subsequent predicted key state. Thus, the **HIS** metric measures similarity without being affected by the number of discovered key states.

## 3.2 MAIN RESULTS

**Baselines.** We first consider several popular baselines without Chain-of-Thought key state prediction: Vanilla BC (Torabi et al., 2018), Decision Transformer (Chen et al., 2021), Behavior Transformer (Shafiullah et al., 2022), MaskDP (Liu et al. 2022, use ground truth key states differently), Decision Diffuser (Ajay et al., 2022). We further consider the baselines with different key state labeling techniques (all these baselines are trained with the CoTPC framework; see Sec. B for more training details): (1) **Last State** only includes the very last state of the trajectory as the key state. (2) **AWE** (Shi et al., 2023) (3) **LLM+CLIP** (Di Palo et al., 2023) first uses Large Language Model (LLM) to generate subgoals for a task, then uses CLIP to target the image frame scoring highest regarding each subgoal, which will serve as the key states. (4) **GT Key States** uses ground truth key states, which is the oracle as in (Jia et al., 2023). Here, labeling methods (1), (3), and (4) can be regarded as zero-shot, which are dependent on human semantics, while methods (2) and InfoCon are intrinsically linked to the inherent properties of the data.

**Performance of InfoCon.** We report the success rate of all baselines in Tab. 1 (additional results of the methods without Chain-of-Thought key state prediction, except Decision Transformer, can be found in Sec. C since their performance is much worse than Decision Transformer according to Jia et al. 2023). Also, as reported in Jia et al. (2023), some of the methods above, which do not use Chain-of-Thought key state prediction strategies, cannot even achieve reasonably good results in seen environments during training (overfitting). Thus, we follow CoTPC to comprehensively compare different baselines and report results on the seen and unseen environment seeds. As observed in Tab. 1, the policies trained with our discovered manipulation key states can achieve competitive performance compared with CoTPC policies using ground-truth key states and other key state grounding methods. Notably, the generalization of InfoCon is also evidenced by delivering top success rates across different tasks. We further provide human intuition similarity of key states grounded by LLM+CLIP, discovered by AWE and InfoCon in Tab. 2. We find that the **HIS** score seems to have a weak correlation with the policy performance presented in Tab. 1. Therefore, we treat this as a signal that the effectiveness of the manipulation concepts may not align well with what human semantics endorse, which justifies the need to develop discovery and grounding methods that can learn from unannotated trajectories.

---

[1]Policies with key states labeled by AWE and LLM+CLIP are trained by us. Other results are from (Jia et al., 2023).

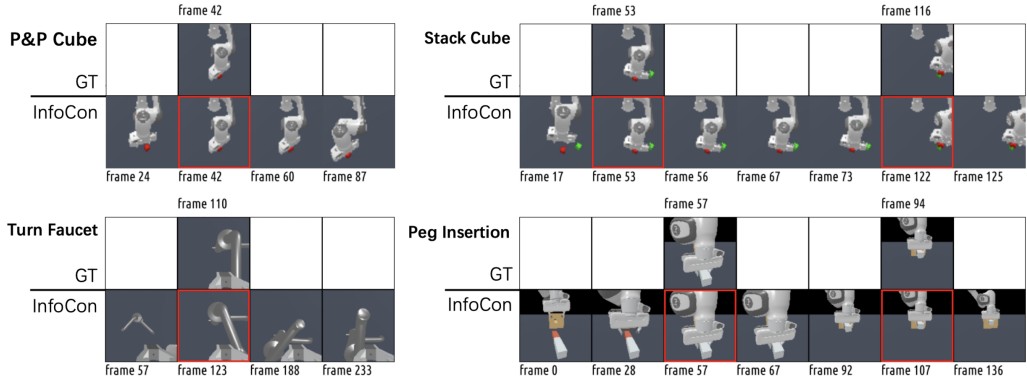

Figure 5: Key states discovered and grounded by InfoCon. From top-left to bottom-right are visualizations of key states of tasks: P&P Cube, Stack Cube, Turn Faucet, and Peg Insertion. Each subfigure contains frames of ground-truth key states at the upper part and key states discovered by InfoCon below. We align frames of ground-truth key states with the nearest subsequent key states from InfoCon by checking their timesteps.

**Visualization of key states.** We visualize the discovered manipulation concepts and grounded key states with InfoCon in Fig. 5. As shown, the identified key states consist of states similar to the ground-truth key states from Jia et al. (2023) (Fig. 4) as well as more fine-grained ones. We provide additional visualizations in Sec. D. We also perform an experiment to assign discovered concepts with semantically meaningful descriptions for the Peg-Insertion task in Sec. E.

### 3.3 ABLATION STUDY

We perform an ablation study to investigate the characteristics of both generative and discriminative goals. Keeping other hyper-parameters constant, we separately omit the loss terms associated with the discriminative goal (Eq. 7 and Eq. 8) and the loss terms related to the generative goal (Eq. 10). Our evaluation considers two primary metrics: the success rate of CoTPC policies on the four manipulation tasks and the Human Intuition Similarity (**HIS**, Eq. 11) on those tasks.

Table 3: Ablation study of the policy performance with only generative loss or discriminative losses. Here, "All" means the InfoCon trained with both the generative and discriminative goal losses.

| SR(%) | P&P Cube | | Stack Cube | | Turn Faucet | | | Peg Insertion | |
|---|---|---|---|---|---|---|---|---|---|
| | seen | unseen | seen | unseen | s.&sf. | u.&sf. | u.&uf. | seen | unseen |
| All | **96.6** | **81.0** | **63.0** | **47.0** | **53.8** | **52.0** | 17.8 | **63.6** | **17.8** |
| Generative Goal Only | 69.8 | 61.0 | 46.8 | 25.0 | 34.8 | 31.0 | **19.8** | 50.2 | 5.5 |
| Discriminative Goal Only | 72.0 | 62.0 | 57.4 | 23.0 | 39.2 | 43.0 | 17.8 | **64.2** | 10.5 |

The success rates of different variants are reported in Tab. 3. When only employing the generative loss or the discriminative losses, the policies tend to be worse than those trained with states discovered by both types of losses [2]. While some models might perform well in seen environments, they often fail to deliver comparable results in unseen scenarios. This reaffirms the pivotal role of proper key state identification and underscores the indispensable synergy between the generative and discriminative informativeness losses.

Table 4: Ablation study with the human intuition similarity for either the generative or discriminative losses. Here, "All" means the InfoCon trained with both generative and discriminative losses.

| HIS | P&P Cube | Stack Cube | Turn Faucet | Peg Insertion |
|---|---|---|---|---|
| All | **2.60** | 8.98 | **18.03** | **9.05** |
| Generative Goal Only | 11.15 | 23.11 | 38.79 | 37.11 |
| Discriminative Goal Only | 3.40 | **6.06** | 28.55 | 11.05 |

---

[2]In this context, we select the best policies achievable under the ablated configurations

The ablations with **HIS** are reported in Tab. 4. When employing only the generative loss, the identified key states exhibit reduced similarity to human intuition. Interestingly, the effect on intuition similarity appears less prominent when relying solely on the discriminative goal loss. We postulate that this might be attributed to the noise in the manipulation concept assignment during the initial training phase. Particularly, at the beginning of training, the groundings of key states are not always accurate. The key states may not faithfully represent goal states, rendering them suboptimal as prediction targets in the generative goal loss (Eq. 6). Practically, during the initial training phase, we can temporally turn off the optimization of the generative goal loss (Eq. 6) and let the discriminative goal loss run for a few epochs. More details on this part can be found in Sec. B.2.

## 4 RELATED WORK

**Behavior Cloning in Manipulation Tasks.** Training robots through demonstration-driven behavior cloning continues to be a key research area (Argall et al., 2009). Within this realm, various techniques have evolved to harness the potential of imitation learning (Zhang et al., 2018; Rahmatizadeh et al., 2018; Fang et al., 2019). Approaches range from employing clear-cut strategies (Zeng et al., 2021; Shridhar et al., 2022; Qin et al., 2022) and more subtle methods to adopting models based on diffusion principles (Chi et al., 2023; Pearce et al., 2023). Some methodologies manage to sidestep the necessity of task-specific labels throughout training and discover robot skills (Shankar & Gupta, 2020; Tanneberg et al., 2021; Xu et al., 2023). Different from prior research, our work prioritizes the autonomous extraction and understanding of manipulation concepts from unlabeled demonstrations, taking a step beyond mere replication toward genuine conceptual understanding.

**Hierarchical Planning and Manipulation Concept Discovery.** Hierarchical planning is crucial in interaction and manipulation scenarios, as it underscores the need for blending broad strategies with specific, detailed actions to achieve optimal results (Hutsebaut-Buysse et al., 2022; Yang et al., 2022; Xu et al., 2018; Jia et al., 2023). Recent years, the Chain of Thought (CoT) prompting methodology, as depicted in (Wei et al., 2022; Cheng et al., 2022; Yao et al., 2022), further emphasizes the value of breaking down intricate tasks into a succession of more straightforward sub-tasks. By doing so, the complexity of the policies is significantly reduced, paving the way for simpler learning processes and resulting in more reliable policies. However, most hierarchical planning methods lean heavily on human semantic intuition. This can manifest as direct human input or systems mimicking human reasoning like LLMs (Di Palo et al., 2023). Some studies (Zambelli et al., 2021; von Hartz et al., 2022; Sermanet et al., 2018; Morgan et al., 2021; Yan et al., 2020; Weng et al., 2023) employ self-supervised discovery of manipulation concepts to circumvent such manual semantic interventions, leveraging mutual information (Hausman et al., 2018; Gregor et al., 2016), significant points of time (Neitz et al., 2018; Jayaraman et al., 2018; Pertsch et al., 2020; Zhu et al., 2022; Caldarelli et al., 2022), and geometry (Shi et al., 2023; Morgan et al., 2021; Zhu et al., 2022) or physics (Yan et al., 2020) constraints. Recent method AWE (Shi et al., 2023) has been proposed to mitigate the compounding error problem inherent to behavioral cloning and automatically determine waypoints based on trajectory segments that can be approximated by linear motion. However, the interpretability of discovered manipulation skills in these methods remains inadequate. Our proposed InfoCon broadens the scope beyond mere planning or trajectory partitioning, with a deeper understanding and abstraction of underlying manipulation concepts. InfoCon ensures not just task completion but also a richer conceptual grasp of the task itself.

## 5 CONCLUSION

We propose InfoCon, a self-supervised learning method capable of discovering task-specific manipulation concepts based on generative and discriminative informativeness. The concepts discovered by InfoCon share similarities with human intuition and are able to be utilized for training manipulation policies with comparable performance to oracles. Our work provides an idea of constructing embodied agents discovering concepts themselves other than struggling with the grounding of concepts that are manually specified. We hope it is received as an attempt to develop an automatic mechanism for discovering useful abstract concepts. In the future, we will consider exploring the possibility of methods for discovering relationships between manipulation concepts and forming structures of discovered concepts.

## 6 ACKNOWLEDGMENT

This work is supported by the HKU-100 Award, donation from the Musketeers Foundation, the Microsoft Accelerate Foundation Models Research Program, and in part by the JC STEM Lab of Robotics for Soft Materials funded by The Hong Kong Jockey Club Charities Trust. We also like to thank Qihang Fang for helping with the human motion concept discovery experiments.

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

## A  MORE DETAILS OF ARCHITECTURE

### A.1  TIME-STEP EMBEDDING.

For the feature $z$ extracted (Eq. 4) from state $s$ at time-step $t$, we always normalize $z$ to proceed the latter concept assignment (Eq. 5). In Eucild Space, $z$ is on a high dimensional unit ball. In order to maintain this "spherical" characteristic and embed $z$ using time-steps, we design the time-step embedding method below:

$$z \leftarrow [\sin \frac{(2t/T - 1)\pi}{2 + 2A}, z \cdot \cos \frac{(2t/T - 1)\pi}{2 + 2A}]$$
$$1 \leq t \leq T \tag{12}$$

where $A > 0$ and $T$ is the total number of time-steps.

We give an intuition explanation at Fig. 6. This embedding operation will influence more on $z$ with time step $t$ close to the beginning and the end of the manipulation trajectory, making it focus more on the absolute time-step (original weight in $z$ will be relatively small), while for $z$ with time-steps in the middle, they will receive weaker influence from this embedding operation. Reflecting on our intuition, when learning a manipulation task, the information of absolute passing time is informative at the beginning and the end, but in the middle we will focus more on relative time-step order instead of absolute current time.

In Eq. 12 $A$ is a positive hyper parameter controlling the usage of area of the whole spherical surface. When $A = 0$, the whole spherical surface will be used for time step embedding, but when the unified time-step $\frac{t}{T}$ of $z$ is 0 or 1, the embedded vector will always be $(-1, \vec{0})$ or $(1, \vec{0})$, which will annihilate the feature of original vector. So we use coefficient $A$ to avoid this. The inclusion of $A$ will still maintain the characteristic of focusing more on absolute time at the beginning or the end.

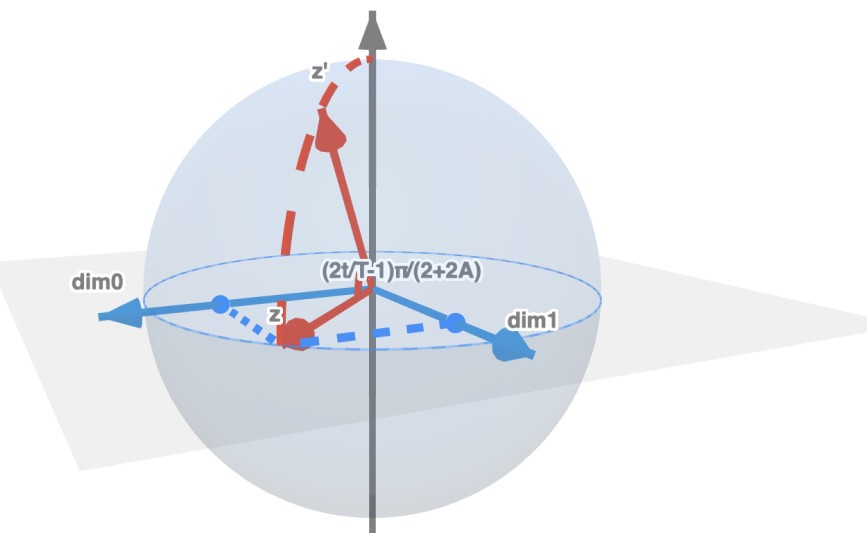

Figure 6: An example of our time-step embedding method. Here we use an example of 2 dimension $z = (x, y)$. When $z$ is at time-step $t$, the embedded feature vector is $z' = (\sin \frac{(2t/T-1)\pi}{2+2A}, x \cos \frac{(2t/T-1)\pi}{2+2A}, y \cos \frac{(2t/T-1)\pi}{2+2A})$.

This embedding method can also restrict the upper bound of cosine similarity between two feature vector when time step is different:

$$\begin{aligned}
&\langle [\sin t_1, z_1 \cos t_1], [\sin t_2, z_2 \cos t_2] \rangle \\
&= \sin t_1 \sin t_2 + \langle z_1, z_2 \rangle \cos t_1 \cos t_2 \\
&\leq \sin t_1 \sin t_2 + \cos t_1 \cos t_2 \\
&= \cos(t_1 - t_2)
\end{aligned} \tag{13}$$

Here $t_1, t_2 \in [-\frac{\pi}{2}, \frac{\pi}{2}]$ when using the embedding method in Eq. 12. $\langle \cdot, \cdot \rangle$ is cos similarity (since the vectors here are all unit vectors, it is same as inner product). Larger difference between $t_1$ and $t_2$ leads to smaller value of $\cos(t_1 - t_2)$, which means smaller cosine similarity.

## A.2 WHY VQ-VAE?

When discovering key states, we need to give every state in the trajectory a label and decide the key states based on the labels. VQ-VAE naturally provides a process of assigning symbols for inputs, which is suitable for partition and segmentation. The remaining task involves assigning meanings to the vectors in the codebook using self-supervised learning, which is achieved through the proposed generative and discriminative goal losses.

## A.3 DETAILS OF CONCEPT

The concepts $\{\alpha_k\}_{k=1}^K$ is detailed model as two vectors, one for generative goal and one for discriminative goal:

$$\{(\alpha_k, p_k)\}_{k=1}^K$$

$\alpha_k$ will be used for concept assignment (Eq. 5) and prediction of states that achieve the goal in generative goal loss (Eq. 6), and $p_k$ is the compressed parameters for compatibility function (Eq. 2). $p_k$ is able to be transformed into a simple MLP network using hyper-network (David et al. 2016, detailed design in Sec. A.7). See Sec. A.4, Sec. A.5 also for more details of usage and update of the parameters in concepts during training.

## A.4 UPDATE OF PROTOTYPES

We use EMA moving (Li et al., 2020) to update the prototype of features representing concepts. when training and $\alpha_k$ is assigned with a set of extracted features (Eq. 4). We will update $\alpha_k$ using the average of the extracted features:

$$\bar{z} = \text{Normalize}\left(\frac{1}{|\{z : \eta(z) = k\}|} \sum_{\eta(z)=k} z\right)$$

$$\alpha_k \leftarrow \text{Normalize}(c_{ema}\alpha_k + (1 - c_{ema})\bar{z}) \tag{14}$$

Here $0 < c_{ema} < 1$. Notice that we did not update $\alpha_k$ based on each single feature $z$, since we find it is inefficient when training. Our experiments use $c_{ema} = 0.9$.

## A.5 PRESERVE GRADIENT FLOW

When assigning concepts (Eq. 5), the gradient cannot naturally propagate back to the encoder (Eq. 4). We use technique similar to VQ-VAE to achieve this. If we have a set of concept features $\{\alpha_k\}_{k=1}^K$, and the extracted feature from current state is $z$. We calculate the probability of choosing a certain concept using cosine similarity:

$$p(z \rightarrow \alpha_k) = \frac{\exp(\langle z, \alpha_k \rangle / \tau)}{\sum_{k=1}^K \exp(\langle z, \alpha_k \rangle / \tau)}, \tag{15}$$

So we can use the soft version to preserve gradient:

$$\alpha^{soft} = \sum_{k=1}^K p(z \rightarrow \alpha_k) \text{SG}(\alpha_k) \quad, \quad \alpha_{\eta(z)} = \alpha^{soft} + \text{SG}(\alpha_{\eta(z)} - \alpha^{soft})$$

$$p^{soft} = \sum_{k=1}^K p(z \rightarrow \alpha_k) \text{SG}(p_k) \quad, \quad p_{\eta(z)} = p^{soft} + \text{SG}(p_{\eta(z)} - p^{soft}) \tag{16}$$

Where $\eta(z)$ is the same concept assignment function as in Eq. 5, $\text{SG}(\cdot)$ is the stop gradient operation, and the definition of $\alpha_k$ and $p_k$ is same as concept vectors in Sec. A.4. We only hope the gradient to adjust the **selection** of concept, so we stop the gradient of prototype $\alpha_k$ and compressed parameters $p_k$. $\alpha_k$ will be updated using the gradient from generative goal loss (Eq. 6), EMA (Sec. A.4), and reconstruction regularization at next Sec. A.6 .$p_k$ will be updated using the gradient from discriminative goal loss (Eq. 7 and Eq. 8).

### A.6 RECONSTRUCTION REGULARIZATION

To prevent over smoothing of discovered concepts (always one concept) similar to VQ-VAE, we add in a reconstruction process from extracted features $\alpha$ to original states: $\hat{s}_t^i = \phi^{-1}(\alpha_t^i|\{\alpha_j^i\}_{j=1}^{t-1})$. (Here $\alpha_t^i$ is the assigned concept from $\{\alpha_k\}_{k=1}^K$ to state $s_t^i$ in $\tau_i = \{(s_t^i, a_t^i)\}_{t=1}^{T(i)}$ from data). Since the maximum number of concepts is an hyper-parameter and we would not choose a very large number, we do not hope the reconstruction process to be trained well enough. The training loss for reconstruction:

$$\mathcal{L}^{\text{rec}}(\phi^{-1}) = \frac{1}{N}\sum_{i=1}^{N}\frac{1}{T(i)}\sum_{t=1}^{T(i)}\|s_t^i - \phi^{-1}(\alpha_t^i|\{\alpha_j^i\}_{j=1}^{t-1})\| \tag{17}$$

We "pretrain" InfoCon with only the reconstruction loss above and the classification entropy loss (Eq. 9) before using generative and discriminative goal loss:

$$\mathcal{L}^{\text{pre}} = \mathcal{L}^{\text{rec}}(\phi^{-1}) + \lambda\mathcal{L}^{\text{ent}}(\boldsymbol{\alpha}, \phi) \tag{18}$$

During experiments, we find that the above loss can be used to warm up the concept discovery VQ-VAE structure via pertaining to provide a good initialization. The initialization can then help achieve better convergence when the proposed generative and discriminative losses are employed for self-supervised concept discovery. To show the effectiveness of the initialization above, we provide counts of activated manipulation concepts (within the codebook of the VQ-VAE) with and without the usage of this initialization when discovering key concepts. Our observation is that the above initialization can help discover more fined-grained manipulation concepts, as shown by the number of activated concepts. Detailed results are in Tab. 5 below.

Table 5: The counts of activated manipulation concepts (within the VQ-VAE codebook) of InfoCon when it is trained with (w) and without (w/o) the usage of the initialization in Eq. 18

| | P&P Cube | Stack Cube | Turn Faucet | Peg Insertion |
|---|---|---|---|---|
| w rec. | 4.5±0.5 | 7.0±2.0 | 5.0±2.0 | 7.5±2.5 |
| w/o rec. | 1.5±0.5 | 2.0±1.0 | 2.5±1.5 | 1.5±0.5 |

Notice that there are also other methods that can alleviate over-smoothing in VQ-VAE (Roy et al., 2018), but these methods focus on the geometry clustering characteristic of VQ-VAE, which may conflict with our design of generative goal and discriminative goal. On the other hand, regularization with construction is empirically reasonable according to self-supervised learning.

### A.7 HYPER-NETWORK

We provide details of our hyper-network for compatibility function (Eq. 2) at Fig. 7.

## B TRAINING DETAILS

### B.1 PSEDUO CODE

See 1 for the training scheme of our method.

### B.2 HYPER PARAMETERS

**InfoCon.** We use the structure of Transformer used in (Jia et al., 2023), which refers the design of (Brown et al., 2020). The state encoder (Eq. 4) and state re-constructor in A.6 both use a 4-layer causal Transformer. The goal based policy in Eq. 8 use a 1-layer causal Transformer. The predictor for generative goal in Eq. 6 use a 2-layer causal Transformer. For hyper-network in A.7, we also use only one hidden layer in the generated goal function. The number of concepts is fixed, maximum number of 10 manipulation concepts for all the tasks. The temperature $\tau$ in Eq. 5 and Fig. 7 is 0.1. $A$ in Eq. 12 is 0.2. All the size of hidden features output by Transformers and concept features $\{\alpha_k\}_{k=1}^K$ is 128. When training, the coefficient $\lambda$, $\lambda^{\text{rec}}$ in Eq. 18, Eq. 10 is 0.001, 0.1, and we will

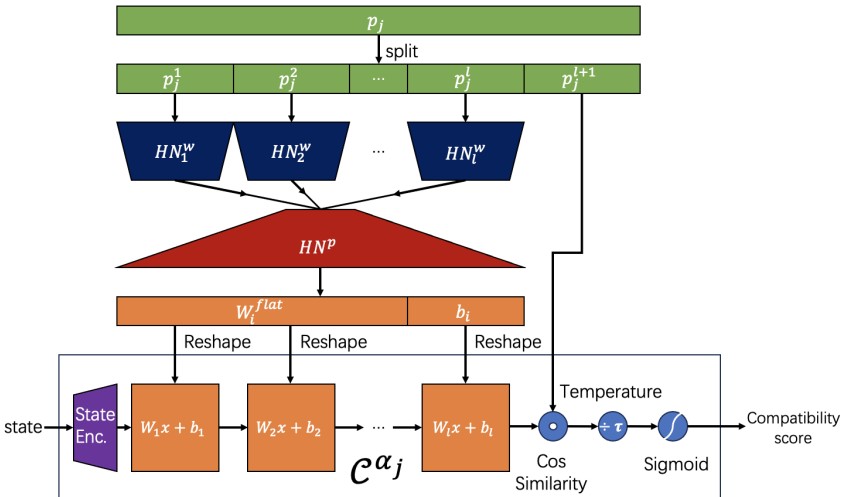

Figure 7: Structure of our hyper-network for compatibility function (Eq. 2). The hyper-network generate hidden layers in the compatibility function, with same shape of weight and bias. We use a two layer hyper-network structure. First set of layers ($HN_i^w$) transform different segments of compressed parameter vectors into medium weight vectors. The weight vectors then pass through the same layer ($HN^p$) and become concatenated flattened weight matrix ($W_i^{flat}$) and bias ($b_i$) for different hidden layers in the compatibility function. The compatibility score will be represent as a sigmoid-activated value of the cosine similarity between the final hidden feature and the last segment of the compressed parameter vector (we will weight them by the temperature $\tau$, which is equal to $\tau$ in Eq. 5).

defer optimization of $\mathcal{L}^{\text{gen}}$ until half of iteration for training is done. We pretrain InfoCon according to Eq. 18 for $1 \times 10^4$ iteration with base learning rate $1.0 \times 10^{-4}$. Then we train the InfoCon for each of the task with $1.6 \times 10^6$ iterations based on the pretrained model with base learning rate $1.0 \times 10^{-4}$. After labeling the original data with key states using trained InfoCon models, we train our CoTPC policies for $1.8 \times 10^6$ iterations with base learning rate $5.0 \times 10^{-4}$. For the three training stages, we all use AdamW optimizer and warm-up cosine annealing scheduler which linearly increases learning rate from $0.1$ of base learning rate to the base learning rate for $1000$ iteration, and then decreases learning rate from base learning rate to $0.1$ of base learning rate. The weight decay is always $1.0 \times 10^{-3}$, and batch size is $256$. For practice, we would only use a segment of $60$ states (along with actions) for every item (trajectory) in the batch.

**Baselines.** Here we only give some essential implementation details of the two baseline methods: CoTPC with LLM sub-goal and CLIP detection of key states, CoTPC with key states discovered by AWE Shi et al. (2023). Notice that after the discover of key states, the training of CoTPC policies are same as description in B.2.

- **CoTPC (LLM+CLIP)**. We discover that the CLIP scores of different key states in a manipulation trajectory are close to each other. Thus, it is hard for us to set a reasonable threshold to decide whether some of the states have already achieved the sub-goal. for each text description of the key states, we will use the average of the minimum and maximum value of CLIP scores to decide the threshold, and select the state with minimum temporal step and is larger than this threshold to be the key states of this sub-goal. (Notice that it is not reasonable to select the maximum score, since the states after achieving the goal still have the feature in the description of key states. Like "grasp the cube", after this sub-goal is achieved, most of the states after it are also suitable for this description in tasks like P&P Cube and Stack Cube.).

- **CoTPC (AWE)**. In AWE research, they set thresholds for end condition of the dynamic programming of finding way-points with different manipulation tasks. Here we modify the method so that it can discover a fix number (here we choose $10$ to align with InfoCon) of key states for all trajectories, which is more suitable since current implementation of CoTPC needs to fix the number of key states.

---

**Algorithm 1** InfoCon

**Input**:      demo trajectories $D_\tau = \{\tau_i = (s_t^i, a_t^i)_{t=1}^{T(i)}\}_{i=1}^N$
**Modules**:    state encoder $\phi$, achieved state predictor $\theta^g$,
               compatilbility function hyper-network $\mathcal{C}$, discriminative goal policy $\pi$
               concepts $\{\alpha_k\}_{k=1}^K$
**Output**:     trained state encoder $\phi$

---

**for** $i = 1, 2, ..., N, t = 1, 2, ..., T(i)$ **do**
     $z_t^i = \phi(s_t^i|\{(s_j^i, a_j^i)\}_{j=1}^{t-1})$
     **for** $k = 1, 2, \ldots, K$ **do**
         $$\mathrm{p}(z_t^i \rightarrow \alpha_k) = \frac{\exp(\langle z_t^i, \mathrm{SG}(\alpha_k)\rangle/\tau)}{\sum_{k=1}^K \exp(\langle z_t^i, \mathrm{SG}(\alpha_k)\rangle/\tau)}$$
     **end for**
     $\alpha_t^i = \alpha_{\arg\max_k p(z_t^i \rightarrow \alpha_k)}$
**end for**
**for** $k = 1, 2, ..., K$ **do**
     $\bar{z}_k = \mathrm{Normalize}(\frac{1}{|\alpha_t^i : \alpha_t^i = \alpha_k|} \sum_{\alpha_t^i = \alpha_k} z_t^i)$
     $\alpha_k \leftarrow \mathrm{Normalize}(c_{ema}\alpha_k + (1 - c_{ema})\bar{z}_k)$
**end for**
**for** $i = 1, 2, ..., N, t = 1, 2, ..., T(i)$ **do**
     **for** $k = 1, 2, \ldots, K$ **do**
         $$\mathrm{p}(z_t^i \rightarrow \alpha_k) = \frac{\exp(\langle z_t^i, \mathrm{SG}(\alpha_k)\rangle/\tau)}{\sum_{k=1}^K \exp(\langle z_t^i, \mathrm{SG}(\alpha_k)\rangle/\tau)}$$       $\triangleright$ Recalculate with updated $\alpha_k$.
     **end for**
     $s_{k_t}^i = s_{\arg\min_u\{u \geq t, \alpha_u^i \neq \alpha_{u+1}^i\}}^i$       $\triangleright$ Last state of sub-trajectory as key state
**end for**
Calculate $\mathcal{L}^{\mathrm{gen}}(\boldsymbol{\alpha}, \phi; \theta^g)$, $\mathcal{L}_c^{\mathrm{dis}}(\boldsymbol{\alpha}, \phi; \mathcal{C})$, $\mathcal{L}_a^{\mathrm{dis}}(\boldsymbol{\alpha}, \phi; \mathcal{C}, \pi)$, $\mathcal{L}^{\mathrm{ent}}(\boldsymbol{\alpha}, \phi)$, $\mathcal{L}^{\mathrm{rec}}(\phi^{-1})$
                                          $\triangleright$ Eq. 6, Eq. 7, Eq. 8, Eq. 9, Eq. 17
$\mathcal{L} = \mathcal{L}^{\mathrm{gen}} + \mathcal{L}_a^{\mathrm{dis}} + \lambda(\mathcal{L}_c^{\mathrm{dis}} + \mathcal{L}^{\mathrm{ent}}) + \lambda^{\mathrm{rec}}\mathcal{L}^{\mathrm{rec}}$
Back propagation from $\mathcal{L}$

---

## C   MORE EXPERIMENT RESULTS

### C.1   OVER-FITTING ISSUE

In Table 1, a noticeable discrepancy is observed in the performance of policies between seen and unseen environments, indicative of an over-fitting issue. Employing the CoTPC with key states of InfoCon, we conducted experiments to evaluate the influence of scaling the training dataset on over-fitting. The results of these experiments are presented in Tab. 6. From these results, it is evident that the impact of augmenting the training data on mitigating over-fitting varies across different tasks.

Table 6: Success rate (%) when varying scale of training set. The policy is CoTPC with key states by InfoCon. For the task Turn-Faucet, we provide results on three kinds of situations: seen environments (s.&sf.), unseen environments with seen faucet types (u.&sf.), and unseen environments with unseen faucet types (u.&uf.).

| SR(%) | P&P Cube | | Stack Cube | | Turn Faucet | | | Peg Insertion | |
|---|---|---|---|---|---|---|---|---|---|
| | seen | unseen | seen | unseen | s.&sf. | u.&sf. | u.&uf. | seen | unseen |
| 200 traj. | 98.5 | 31.0 | 92.0 | 9.0 | 57.0 | 40.0 | 14.0 | 66.0 | 8.5 |
| 500 traj. | 96.6 | 81.0 | 63.0 | 47.0 | 53.8 | 52.0 | 17.8 | 63.6 | 17.8 |
| 800 traj. | 94.4 | 75.0 | 90.6 | 43.0 | 59.2 | 53.0 | 25.3 | 64.3 | 27.0 |

## C.2 More Baseline Methods

We provide the results of baseline methods without the usage of concepts or semantic information in seen environment in Tab. 7. The data is borrowed from (Jia et al., 2023). According to this source, Decision Transformer exhibits the best performance in unseen environments among all methods evaluated. Consequently, the performance of other methods in unseen environments is not provided, except for that of Decision Transformer.

Table 7: More results of success rate (%) of different methods on seen environments.

| SR(%) | P&P Cube | Stack Cube | Turn Faucet | Peg Insertion |
|---|---|---|---|---|
| Vanilla BC | 3.8 | 0.0 | 15.6 | 0.0 |
| Behavior Transformer | 23.6 | 1.6 | 16.0 | 0.8 |
| Decision Diffuser | 11.8 | 0.6 | 53.6 | 0.6 |
| MaskDP | 54.7 | 7.8 | 28.8 | 0.0 |
| Decision Transformer | 65.4 | 13.0 | 39.4 | 5.6 |
| Last State | 70.8 | 12.0 | 47.8 | 9.6 |
| AWE | 96.2 | 45.4 | 53.4 | 31.6 |
| LLM+CLIP | 92.2 | 44.6 | 35.8 | 63.2 |
| GT Key States | 75.2 | 58.8 | 56.4 | 52.8 |
| **InfoCon** | **96.6** | **63.0** | 53.8 | **63.6** |

## C.3 Concepts and concept-driven policies

While our experiments highlight the strength of InfoCon in identifying key states, integrating this with policy generation is an exciting and unexplored area. We see a valuable opportunity for future work in developing a method that not only discovers key states using InfoCon but also generates a policy similar to CoTPC.

Considering the selection of policy, there are not many methods that focus on making use of key states in their decision-making policies with a tailored design. Here are some methods that are relevant to our knowledge:

- **MaskDP** (Liu et al., 2022). This method has a weakness: appending the end state into the input sequence is needed. The agent must know the exact key states based on the initial state, which makes it constrained to achieving a very specific goal and is not applicable to our problem setting, where the end goal state is not provided.

- **Modified Decision Transformer**. This is a method we found in the work of AWE (Shi et al., 2023), in which they adapt the original decision transformer Chen et al., 2021 to let it leverage key state prediction during policy learning.

- **CoTPC** (Jia et al., 2023), which is the method we mainly employ in our work.

We trained the **Modified Decision Transformer** and saw that policies based on InfoCon also have better performance compared with the Decision Transformer without key states or using ground-truth (GT) key states. The results are in the table below.

Table 8: Success rate (%) of methods based on Decision Transformer (DT for short). For the task Turn-Faucet, we provide results on three kinds of situations: seen environments (s.&sf.), unseen environments with seen faucet types (u.&sf.), and unseen environments with unseen faucet types (u.&uf.).

| SR(%) | P&P Cube | | Stack Cube | | Turn Faucet | | | Peg Insertion | |
|---|---|---|---|---|---|---|---|---|---|
| | seen | unseen | seen | unseen | s.&sf. | u.&sf. | u.&uf. | seen | unseen |
| DT | 65.4 | 50.0 | 13.0 | 7.0 | 39.4 | 32.0 | 9.0 | 5.6 | 2.0 |
| DT & GT | 89.8 | 68.0 | 58.4 | 29.0 | 41.2 | 36.0 | 16.0 | 34.6 | 6.3 |
| DT & InfoCon | 95.2 | 70.0 | 62.6 | 32.0 | 48.0 | 48.0 | 17.0 | 52.0 | 7.8 |

## D    MORE VISUALIZATION RESULTS

We provide more visualization results of key states labeled out by InfoCon at Fig.10, 11, 12, 13, 14, 15

## E    ALIGNMENT WITH HUMAN SEMANTICS

A variety of methods have been devised to identify elements similar to key states (Neitz et al., 2018; Jayaraman et al., 2018; Pertsch et al., 2020; Zhu et al., 2022; Caldarelli et al., 2022). Our approach surpasses them by ensuring that the identified key states stem from well-defined concepts. Specifically, in our framework, these concepts pertain to the modeling of goal (Eq. 1,Eq. 2,Eq. 3). Furthermore, we demonstrate that the concepts encapsulated within the key states of InfoCon share similarity with intricate human semantics, as substantiated by the evaluation of the Human Intuition Score (HIS, Eq. 11).

In this section, we present further attempts and results to align the self-discovered manipulation concepts by InfoCon with human semantics for potential enhancement in the explainability when facing human-robot interaction. We perform the analysis with the robotic task of "Peg-Insertion." The major goal is to check whether we can assign reasonable and linguistically meaningful names or descriptions to the discovered concepts or key states.

Before we perform some qualitative analysis, we first determine the manipulation concepts discovered for this task by checking the activated concepts (codes) among all the codes in the codebook (in total 10) of the VQ-VAE employed. Namely, all the trajectories of the task "Peg-Insertion" are pushed through the VQ-VAE, the activated codes will be recorded for each of them, and then the activation rate for each code (concept) is computed.

Tab. 9 provides the activation rate of each manipulation concept related to task Peg Insertion using the concept discovery VQ-VAE trained by InfoCon.

Table 9: Activation rate of each manipulation concept on task Peg Insertion from InfoCon.

| Concept Index | 0 | 1 | 2 | 3 | 4 | 5 | 6 | 7 | 8 | 9 |
|---|---|---|---|---|---|---|---|---|---|---|
| Activation Rate (%) | 46.7 | 65.0 | 100.0 | 1.9 | 91.1 | 100.0 | 95.6 | 82.3 | 100.0 | 3.9 |

The table above indicates that nearly every trajectory performing task Peg Insertion has the same set of key states activated, which means that the key states (concepts) involved in accomplishing Peg Insertion are the discovered concepts with numbering: #0, #1, #2, #4, #5, #6, #7, and #8, in total 8 vectors in the code book of VQ-VAE in InfoCon. (#3 and #9 are probably noise, given their low activation rate).

Next, we use a sequence to illustrate the description we assign to each of the discovered concepts, which help align with human semantics. The concept names are the following:

1 . The gripper is positioned above the peg (discovered concept #7).
2 . The gripper is aligned with the peg and ready to grasp (discovered concept #5).
3 . The peg is grasped (discovered concept #0).
4 . The peg is grasped and lifted (discovered concept #1).
5 . The peg is aligned with the hole distantly (discovered concept #4).
6 . The peg is aligned with the hole closely (discovered concept #6).
7 . The peg is inserted half-way into the hole (discovered concept #8).
8 . The peg is fully inserted (discovered concept #2).

Please check Fig. 8 for visuals corresponding to these discovered concepts. The above verifies that humans can still assign descriptions to the discovered concepts by InfoCon. At the same time, we can see the potential of InfoCon to discover meaningful concepts that can be aligned with human

semantics. Moreover, it shows that InfoCon has the capability of discovering more fine-grained concepts or key states human annotators (in CoTPC) have ignored or are unaware of. These further demonstrate the effectiveness of the proposed InfoCon for automatically discovering meaningful manipulation concepts.

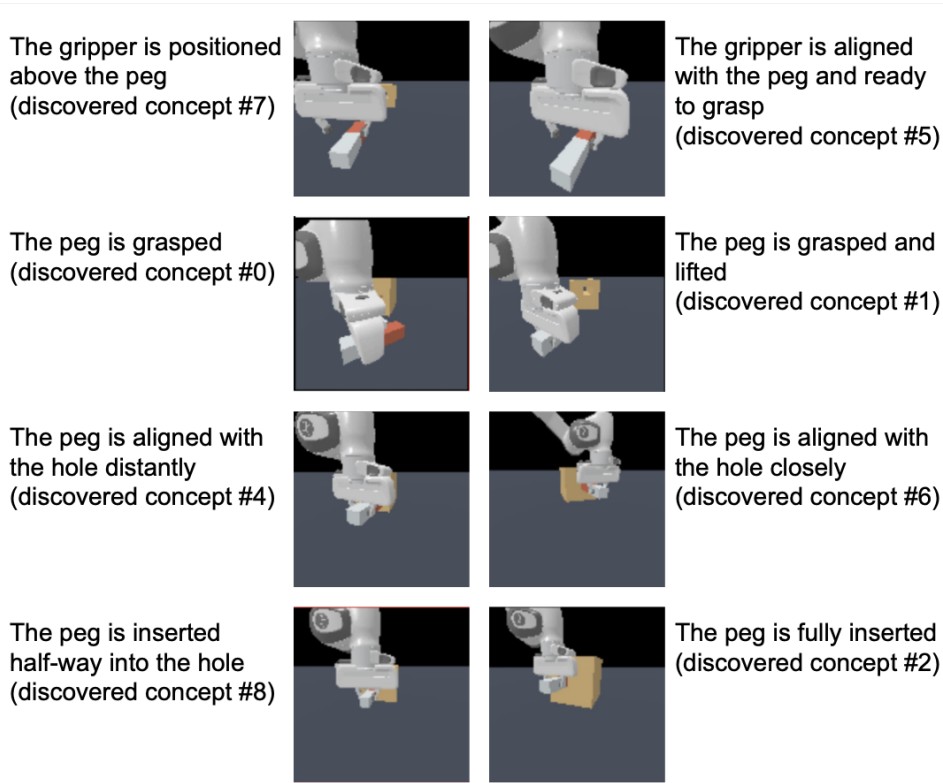

Figure 8: Align InfoCon Concepts with Human Concepts on the Peg Insertion task.

## F  INFOCON WITH REAL-WORLD DATA

InfoCon can be applied in various tasks involving sequential events (from manipulation, to sound, video, natural language, log of computer systems, cache behavior of memory, and changing of climate and weather). These processes can all be regarded as a trajectory with meaningful partitions in achieving specific goals. Here we provide a visualization of the results of applying InfoCon on human pose sequences estimated from real-world videos. We can see that it also separates a video of pose motion into different key states, which can be mapped to different "human motion concepts".

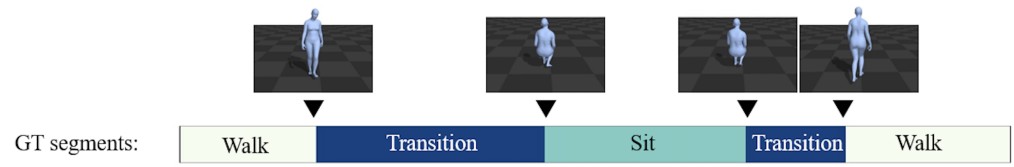

Figure 9: InfoCon for human motion concept discovery from pose sequences estimated with real-world videos.

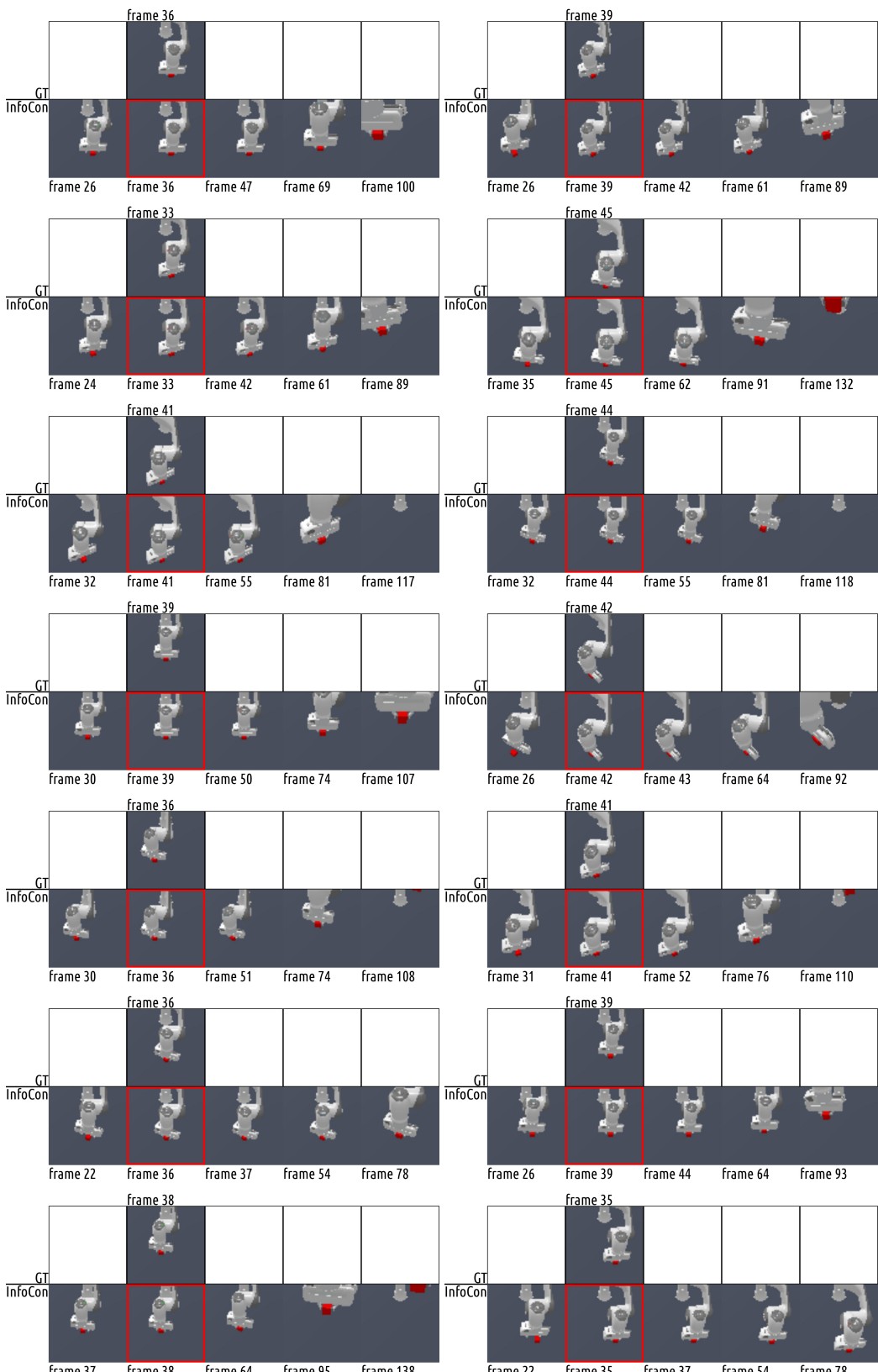

Figure 10: More examples of labeled out key states of task **P&P Cube** (part1).

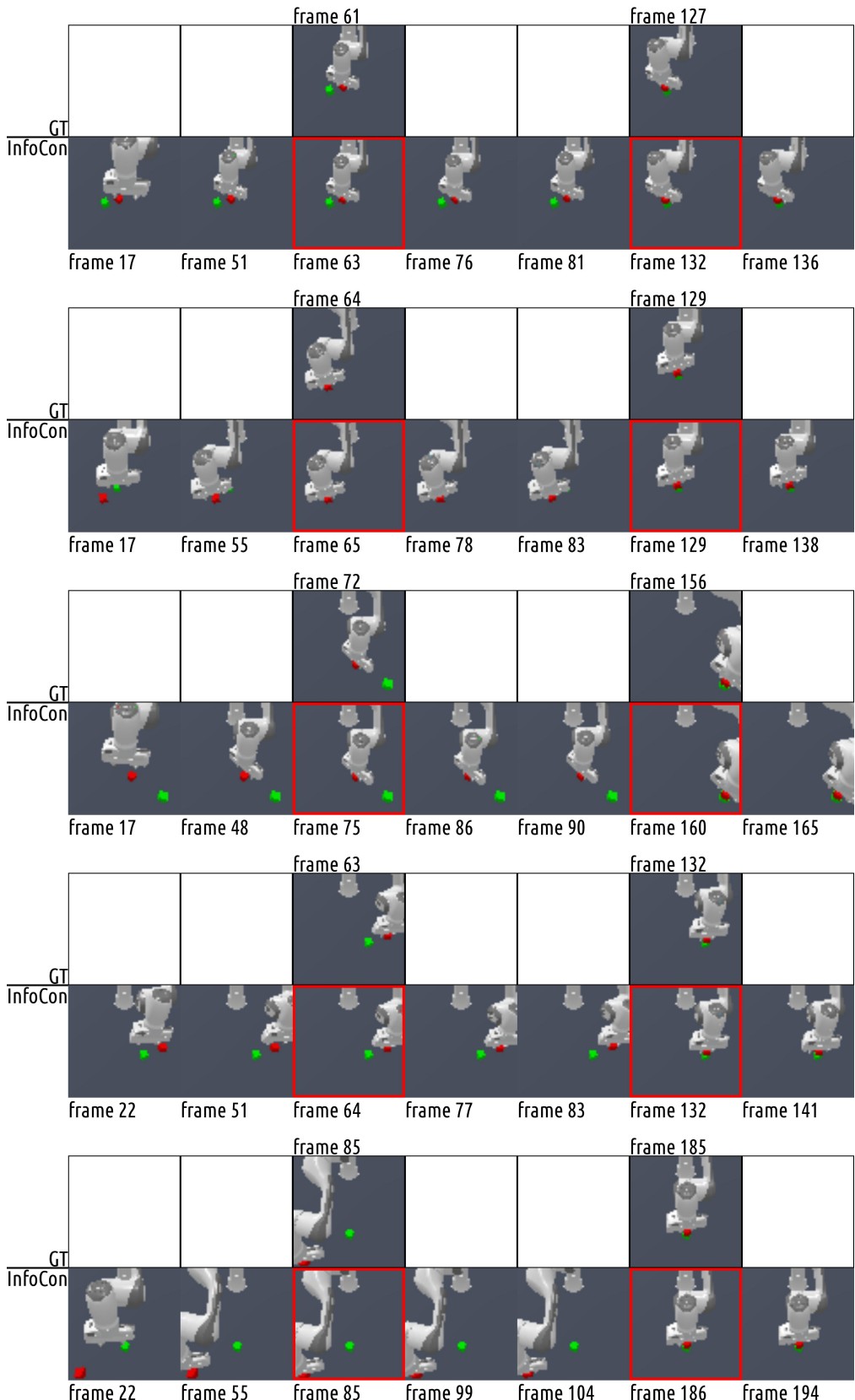

Figure 11: More examples of labeled out key states of task **Stack Cube** (part1).

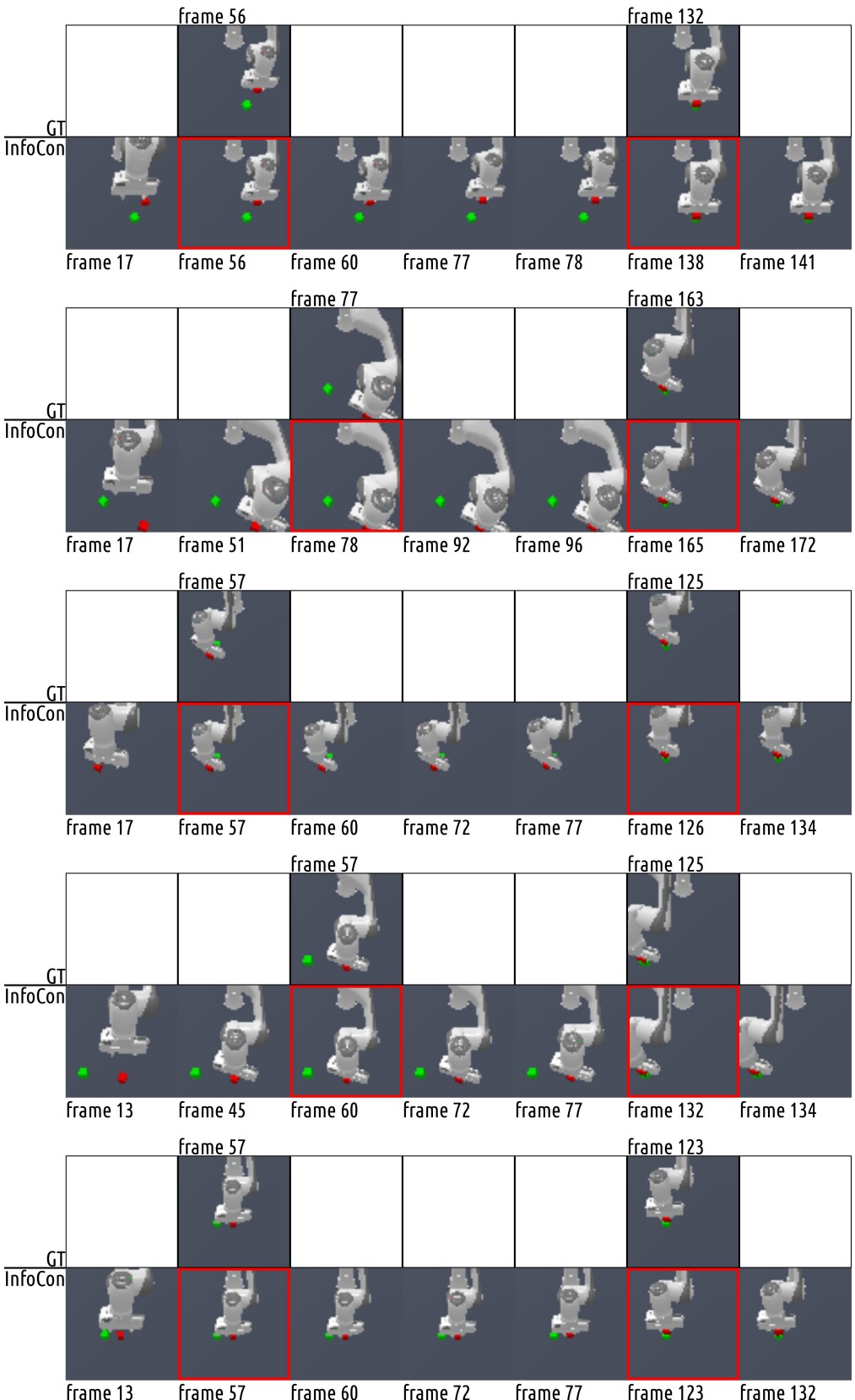

Figure 12: More examples of labeled out key states of task **Stack Cube** (part2).

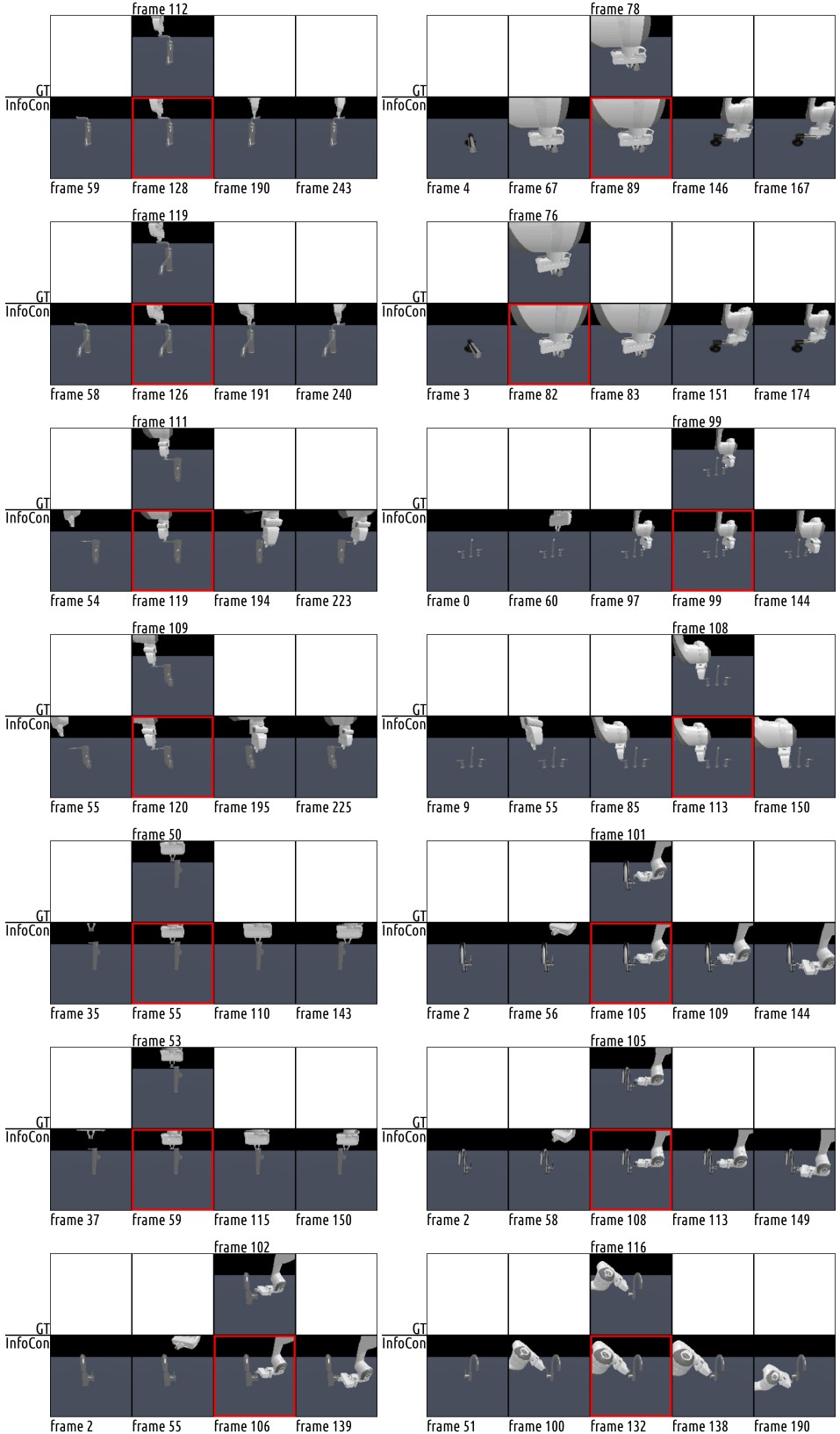

Figure 13: More examples of labeled out key states of task **Turn Faucet**.

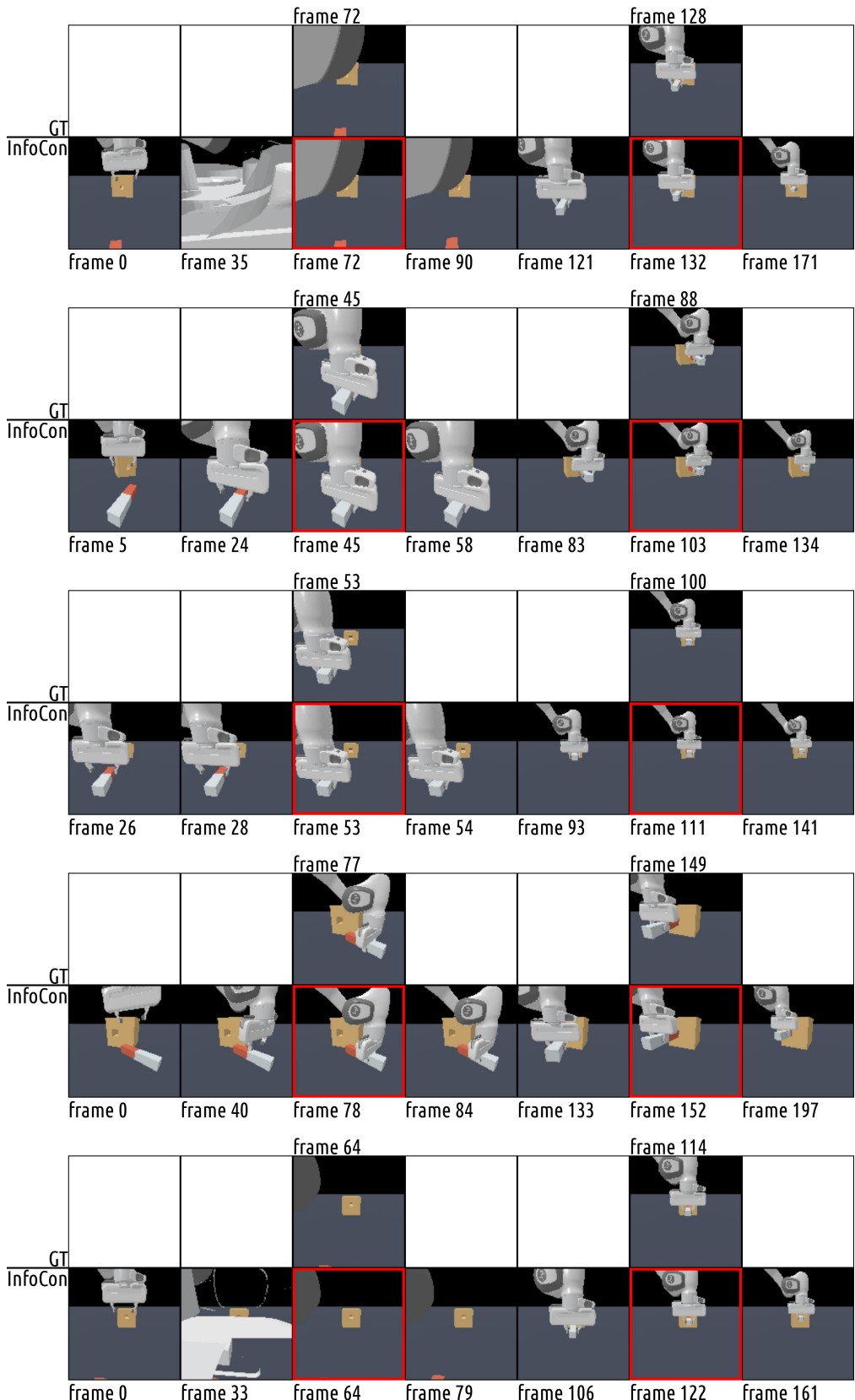

Figure 14: More examples of labeled out key states of task **Peg Insertion** (part1).

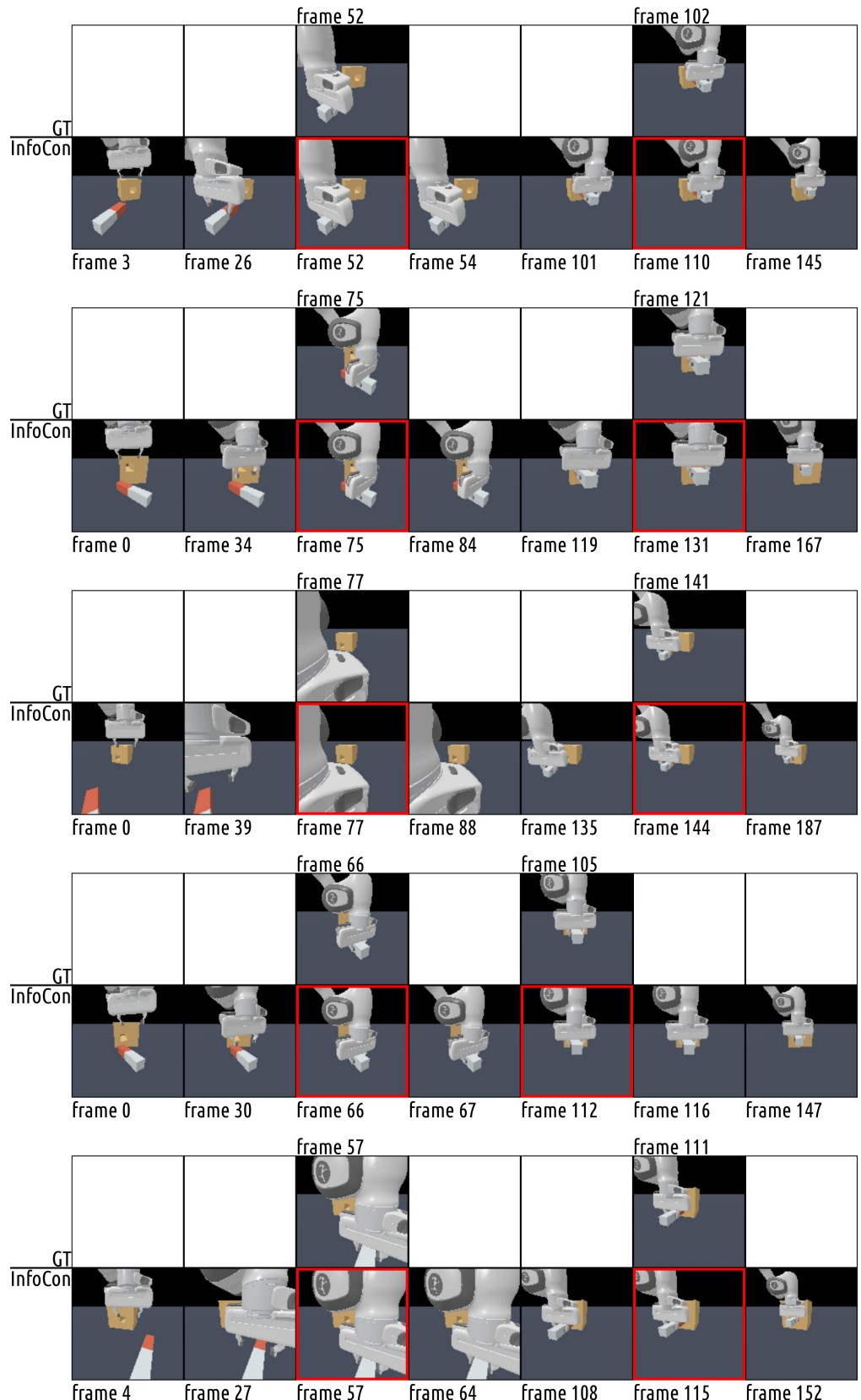

Figure 15: More examples of labeled out key states of task **Peg Insertion** (part2).

