# OpenReview forum: "InfoCon: Concept Discovery with Generative and Discriminative Informativeness"
_ICLR.cc/2024/Conference — ICLR 2024 poster_

### Official Review · Reviewer_v8jc · 2023-10-27

**Soundness:** 2 fair
**Presentation:** 3 good
**Contribution:** 3 good
**Rating:** 8
**Confidence:** 4

**Summary:**

This paper proposes a method for extracting different manipulation concepts (i.e. grasp, align, insert) from expert trajectories of robot manipulation tasks. In essence, the algorithm unsupervisedly learns relevant subgoals to accomplish a task, and in addition, also provides a gradient signal for actions to achieve these (sub)goals. This is done by training two components: predicting the next subgoal given the current state, and training a compatibility function that indicates how "compatible" a state is with the desired subgoal, using contrastive learning. The gradient of the compatibility function can then be used to select actions, i.e. which action increases the goal compatibility given the current state. The method is evaluated on 4 tasks of the ManiSkill2 benchmark, and some qualitative examples are provided of the discovered subgoals.

**Strengths:**

- Interesting ideas and approach to go from expert trajectories to subgoals, and in addition obtain policies to accomplish those subgoals.

**Weaknesses:**

- The experimental results don't provide standard deviations, which makes it difficult to assess if there is any significant improvement compared to the presented baselines.

**Questions:**

- What are the standard deviations on the results in Table 1. To what extent is InfoCon actually significantly better than the other baselines?

- For some tasks (i.e. P&P Cube) the GT key states underperform the other approaches. Any insight on why the ground truth key states are insufficient to efficiently execute the task, and which are the "extra" subgoals identified by InfoCon that might explain this gap?

- The model is trained on only 500 trajectories. To what extent is this overfitting to the train set, and does this explain the large gap between seen and unseen scenarios? Would this gap be closed by just adding more trajectories?

- The VQ-VAE is pretrained on the trajectories without any task-related signal. Hence, the learned subsequences are merely clustered by visual appearance, rather than semantic relevance of being a valid "subgoal" for a task?  Wouldn't it make sense to also adjust the codes in the codebook based on e.g. how good one can predict a particular goal and/or how well-behaved a compatibility function is?

- The policy is conditioned on the current state and the gradient of the compatibility function. Wouldn't it make sense to also condition the policy on the goal state, i.e. similar to e.g. https://arxiv.org/abs/2211.13350

---

> ### Author Response · Authors · 2023-11-21
>
> Dear Reviewer v8jc,
>
> Thank you for your comprehensive review and the valuable insights provided on our paper. We are grateful for your recognition of the innovative aspects of our approach in extracting manipulation concepts from expert trajectories.
>
> We have provided detailed responses to each of your questions. We believe these responses will clarify the points you raised and provide a deeper understanding of our methodology and findings. Especially, on providing deviations, more analysis of the usefulness of the subgoals, ablation with more training demos, and discussion on the implementation details.
>
> We hope that these clarifications can help finalize your assessment and the rating of our paper. Please also let us know if you have any further questions that we need to provide additional clarifications.
>
> ***
>
> **W1, Q1: What are the standard deviations on the results in Table 1. To what extent is InfoCon actually significantly better than the other baselines?**
>
> **A**: Thanks for the question. We provide the statistics of the policy performance when the initial state of each evaluation sample is under perturbations, in order to simulate the noise happening in the real world. As can be seen from the following, the performance is relatively stable compared with our original results.
>
> | InfoCon | P&P Cube | Stack Cube | Turn Faucet | Peg Insertion |
> |:-|:-|:-|:-|:-|
> | Seen Env. | 96.3 ± 0.1 | 63.2 ± 0.4 | 53.5 ± 0.3 | 65.0 ± 1.4 |
> | Unseen Env. | 80.0 ± 1.0 | 46.0 ± 1.0 | 52.0 ± 1.0 (sf.) 17.6 ± 0.6 (uf.) | 17.4 ± 0.4 |
>
> ***
>
> **Q2: For some tasks (i.e. P&P Cube) the GT key states underperform the other approaches. Any insight on why the ground truth key states are insufficient to efficiently execute the task, and which are the "extra" subgoals identified by InfoCon that might explain this gap?**
>
> **A**: Thanks for the insightful comment. We have examined the discovered concepts in Fig. 8 in Sec. E of the updated paper. Please check the details in Sec. E for the experiment with the task of Peg-Insertion.
>
> In contrast to only annotating a few key states (e.g., three, by human-defined rules), InfoCon can actually discover more fine-grained yet semantically meaningful key states (in Peg-Insertion, in total 8 concepts are discovered), which are shown below:
>
> * 1. The gripper is positioned above the peg (discovered concept \#7).
> * 2. The gripper is aligned with the peg and ready to grasp (discovered concept \#5).
> * 3. The peg is grasped (discovered concept \#0).
> * 4. The peg is grasped and lifted (discovered concept \#1).
> * 5. The peg is aligned with the hole distantly (discovered concept \#4).
> * 6. The peg is aligned with the hole closely (discovered concept \#6).
> * 7. The peg is inserted half-way into the hole (discovered concept \#8).
> * 8. The peg is fully inserted (discovered concept \#2).
>
> As observed, InfoCon discovered extra key states, which are also helpful. For example, the extra key state: “The gripper is aligned with the peg and ready to grasp” (item \#2 above). This is a helpful key state that promotes a more stable process to successfully perform the whole peg insertion task (positioning the two handles of the gripper on two sides of the peg can have more guarantee on the next grasp process). This could be the main reason that the discovered concepts can perform better for policy training.
>
> ***
>
> **Q3: The model is trained on only 500 trajectories. To what extent is this overfitting to the train set, and does this explain the large gap between seen and unseen scenarios? Would this gap be closed by just adding more trajectories?**
>
> Below is an experiment of increasing training data of [CoTPC + InfoCon Key states] on task Peg Insertion from 500 to 800. It seems that the performance on unseen environments has improved. But the improvement on seen environments is relatively small. This indicates that the over-fitting gap can be narrowed by using more training data.
>
> || 500 traj. | 800 traj. |
> |:-:|:-:|:-:|
> | Seen Env. | 63.6 | 64.3 |
> | Unseen Env. | 17.8 | 27.0 |
>
> Since all the experiment results we used from CoTPC are trained on 500 trajectories, we keep this number in our experiments for fairness.

---

> > ### Author Response · Authors · 2023-11-21
> >
> > **Q4: The VQ-VAE is pretrained on the trajectories without any task-related signal. Hence, the learned subsequences are merely clustered by visual appearance, rather than semantic relevance of being a valid "subgoal" for a task? Wouldn't it make sense to also adjust the codes in the codebook based on e.g. how good one can predict a particular goal and/or how well-behaved a compatibility function is?**
> >
> > **A**: Training stability and convergence efficiency are the main reasons for utilizing the pre-training with reconstruction and entropy losses.
> >
> > First, without pretraining, the partitioning may not be continuous, which could result in many noisy key states at the beginning. This creates learning instabilities, and it takes time for the training to first converge to a good stage so that the proposed generative and discriminative losses can be optimized more efficiently.
> >
> > Second, through our experiments, we found that a slight pretraining helps InfoCon to discover more fine-grained manipulation concepts that are important for learning policies. The reason is that the reconstruction can help activate more concepts (codes in the codebook) so that they get some initial training signal, which prevents accidentally falling into local minima (e.g., clustering into a very few number of concepts).
> >
> > Our observation is that the role of pretraining goes beyond merely clustering based on appearance, but it has more impact on the learning dynamics. Moreover, the pretraining is performed with much fewer steps than the full training (including all loss terms).
> >
> > Please refer to Sec. B.2 in the appendix for more training details.
> >
> > ***
> >
> > **Q5:  Wouldn't it make sense to also condition the policy on the goal state, i.e. similar to e.g. https://arxiv.org/abs/2211.13350**
> >
> > **A**: The policy employed for discriminative goals is distinct from the one used for downstream tasks (trained with CoTPC policies). We only use the policy for increasing the discriminative informativeness, and do not expect it to practically perform well on downstream tasks.
> >
> > Specifically, the “policy” in Eq.8 that is utilized for minimizing the discriminative goal loss does not include additional information by adding a (sub)goal state. The design is to help maintain the focus on increasing discriminative informativeness, as the training of the policy in Eq.8 may utilize less information from the gradient of the compatibility function when the goal state is available. It also helps with our analysis of the usefulness of the actionable informativeness of the discriminative goal.
> >
> > In addition, we do train policies that leverage (predict) subgoal states (similar to CoTPC) with the discovered key states from InfoCon, and we have shown their efficiency in learning and their generalization to unseen environments.

---

> > > ### Comment · Reviewer_v8jc · 2023-11-22
> > >
> > > I'd like to thank the authors for addressing my questions, and also for providing an extra experiment to improve the unseen case.
> > >
> > > I have increased my score accordingly.

---

> > > > ### Author Response · Authors · 2023-11-22
> > > >
> > > > We are glad you have an updated evaluation of our paper. Thank you again for your review and precious advice!

---

### Official Review · Reviewer_WSFG · 2023-10-31

**Soundness:** 3 good
**Presentation:** 3 good
**Contribution:** 3 good
**Rating:** 6
**Confidence:** 3

**Summary:**

This work, InfoCon, uses self-supervised learning method to discover the manipulation concepts in robotic tasks. The concepts are verified with semantic meaning in terms of human linguistics while saving much manual annotation efforts. This can be used as auxiliary task to support the encoding in the policy optimization. Experiements demonstrate that the policy trained based on these learned concepts can achieve the state-of-the-art results.

**Strengths:**

1. InfoCon can be self-supervised given state-action trajectory without human annotation, guided by network architecture VQ-VAE and informativeness objectives. Surprisingly, the self-supervised key states even performs better than the human GT in the COTPC for policy generation.

2. The robot with InfoCon can discover abstract concepts themselves other than struggling with the grounding of concepts that are manually defined.

3, The concepts of generative goal and discriminative goal are novel and beneficial to the trajectory encoding, which serves as the auxiliary task for self-supervision.

4. Strong results in simulation comparing to extensive baselines.

**Weaknesses:**

1, The proposed approach and concept of self-supervised manipulation concepts and key states seems closely related to the COTPC, as COTPC needs the key states. However, in the method description, there is no mention of COTPC. It may be possible to achieve co-optimization between policy generation and self-supervised manipulation concept. Moreover, can the proposed approach be general beneficial to policy optimization beside COTPC?

2. The experience portion is a bit weak where only few tasks are evaluated. For the baseline, it should also include COTPC + other manipulation concept discovery for fair comparison.

3. Lack of real robotic experiments. It is hard to judge if the proposed self-supervised approach works for the real-world complex tasks and videos.

**Questions:**

1. As generative models, there are many VAE variants. Can you explain why you choose VQ-VAE architecture or include ablation study?

2. As mentioned in the paper: the manipulation concept, key state, and state are random variables depending on the trajectory. Does the initialization of these variables affect the experimental results? Or other prototype network approaches help?

3. How about the generalization capability of InfoCon? In the paper, the training and testing tasks are same: P&P Cube, Stack Cube, Turn Faucet and Peg Insertion.

---

> ### Author Response · Authors · 2023-11-21
>
> Dear Reviewer WSFG,
>
> Thank you for your constructive and insightful feedback on our manuscript. We appreciate your recognition of the novelty and effectiveness of InfoCon, particularly the self-supervised characteristics, its ability to discover manipulation concepts with semantic meaning, and its contribution to enhancing policy learning.
>
> Below, we have provided clarification and extra experimental results to address the comments and questions. Especially on the discussion of the co-optimization, the benefits of the proposed to other policy learning schemes, and some ablations for further understanding of the proposed.
>
> We hope that these clarifications can help finalize your assessment and the rating of our paper. Please also let us know if you have any further questions that we need to provide additional clarifications.
>
> ***
>
> **W1.1: The proposed approach and concept of self-supervised manipulation concepts and key states seems closely related to the COTPC, as COTPC needs the key states. However, in the method description, there is no mention of COTPC.**
>
> **A**: Thanks for the comment. We put the details about CoTPC for training concept-guided policies in the training details section, and we will move this part to the method section in our final version.
>
> ***
>
> **W1.2: It may be possible to achieve co-optimization between policy generation and self-supervised manipulation concept.**
>
> **A**: Thanks for the suggestion. We would like to explore this idea in future research. Currently, we do not perform joint training as the policy training would impose another layer of dynamics that could make it difficult to analyze the efficiency of the proposed generative and discriminative goal losses.
>
> While our experiments highlight InfoCon's strength in identifying key states, integrating this with policy generation is an exciting and unexplored area. We see a valuable opportunity for future work in developing a method that not only discovers key states using InfoCon but also generates a policy similar to CoTPC.
>
> ***
>
> **W1.3: Moreover, can the proposed approach be general beneficial to policy optimization beside COTPC?**
>
> **A**: In this paper, we mainly employ CoTPC for the evaluation of the usefulness of the discovered key states, as currently, there are not many methods that focus on making use of key states in their decision-making policies with a tailored design. Here are some methods that are relevant to our knowledge:
>
> * CoTPC.
> * MaskDP[1]. However, it has a weakness: appending the end state into the input sequence is needed. The agent must know the exact key states based on the initial state, which makes it constrained to achieving a very specific goal and is not applicable to our problem setting, where the end goal state is not provided.
> * From the work of AWE [2], we found that DT performs best, so we present extra experiments on Decision Transformer [2] (DT), where we adapt DT to let it leverage key state prediction during the policy learning (our adaptation is based on the implementation in [3]).
>
> We trained the adapted DT and tested on Peg Insertion and saw that policies based on InfoCon also have better performance compared with DT without key states or using ground truth (GT) key states.
>
> Results are in the table below.
> Peg Insertion | DT | DT+GT | DT+InfoCon |
> |:-:|:-:|:-:|:-:|
> Seen Env. | 5.6 | 34.6 | 52.0 |
> Unseen Env. | 2.0 | 6.3 | 7.8 |
>
> Given this evidence, we believe that InfoCon is beneficial to the usage of key states in different strategies or policies.
>
> [1]. Liu, F., Liu, H., Grover, A., & Abbeel, P. (2022). Masked autoencoding for scalable and generalizable decision making. Advances in Neural Information Processing Systems, 35, 12608-12618.
>
> [2]. Chen, L., Lu, K., Rajeswaran, A., Lee, K., Grover, A., Laskin, M., ... & Mordatch, I. (2021). Decision transformer: Reinforcement learning via sequence modeling. Advances in neural information processing systems, 34, 15084-15097.
>
> [3]. Shi, L. X., Sharma, A., Zhao, T. Z., & Finn, C. (2023). Waypoint-based imitation learning for robotic manipulation. arXiv preprint arXiv:2307.14326.
>
> ***
>
> **W2: it should also include COTPC + other manipulation concept discovery for fair comparison.**
>
> **A**: Thanks for the comment. We have included a few CoPTC+other manipulation concepts baselines (in total 4) in Table 1 of Section 3 (Experiments) in the manuscript (Last State, AWE, LLM + CLIP, GT Key States). We will carry out more experiments on this aspect in the final version.
>
> ***
>
> **W3: Real-world examples**
>
> **A**: Thanks for the comment. Due to the time limit, we have run the proposed InfoCon on human pose sequences estimated from real-world videos to check its efficiency in discovering human motion concepts. A qualitative result is presented in Fig. 9. With this experiment, we show the potential of the proposed InfoCon in discovering key states from real-world data, but we are interested in training InfoCon on large-scale data when time permits.

---

> > ### Author Response · Authors · 2023-11-21
> >
> > **Q1: Can you explain why you choose VQ-VAE architecture or include ablation study?**
> >
> > **A**: When discovering key states, we need to give every state in the trajectory a label and decide the key states based on the labels. VQVAE naturally provides a process of assigning symbols for inputs, which is suitable for partition and segmentation. The remaining task involves assigning meanings to the vectors in the codebook using self-supervised learning, which is achieved through the proposed generative and discriminative goal losses. We‘ve also tried the basic VAE method at the beginning and found that it is hard to separate the trajectories into discrete symbols.
> >
> > ***
> >
> > **Q2: Does the initialization of these variables affect the experimental results? Or other prototype network approaches help?**
> >
> > **A**: Thanks for the insightful comment. Through our experiments, we found that proper initialization of the VQVAE code book helps alleviate the problem of training instability (please refer to Sec A.5 in the updated manuscript).
> >
> > Specifically, we have experimented with the initialization of the VQ-VAE using solely the reconstruction and the entropy loss in Eq. 18. We compare the number of discovered concepts with and without the pretraining. We found that initialization using reconstruction would encourage more discovered concepts as it can help prevent the training from getting stuck in local minima. The detailed experimental procedure can be found in Appendix A.5. The table below summarizes the effect of initializing the VQ-VAE code book using reconstruction. We can see that InfoCon can discover more manipulation concepts when using reconstruction.
> >
> > || P&P Cube | Stack Cube | Turn Faucet | Peg Insertion |
> > |:-:|:-:|:-:|:-:|:-:|
> > | w rec. | 4.5±0.5 | 7.0±2.0 | 5.0±2.0 | 7.5±2.5 |
> > | w/o rec. | 1.5±0.5 | 2.0±1.0 | 2.5±1.5 | 1.5±0.5 |
> >
> > ***
> >
> > **Q3: How about the generalization capability of InfoCon? In the paper, the training and testing tasks are same: P&P Cube, Stack Cube, Turn Faucet and Peg Insertion.**
> >
> > **A**: Thanks for the question. We would like to clarify that the experimental setting (following previous works) actually considers the generalization of the trained policies on unseen scenes of the tasks. In terms of the generalization of the discovery, we have also experimented with human pose sequences estimated from real-world videos. A qualitative result is displayed in Fig. 9, which demonstrates that InfoCon can generalize to a different domain by discovering meaningful concepts in human motion. In the future, we will test InfoCon on many more domains.

---

### Official Review · Reviewer_mBdA · 2023-11-01

**Soundness:** 3 good
**Presentation:** 4 excellent
**Contribution:** 3 good
**Rating:** 6
**Confidence:** 3

**Summary:**

The authors propose using vector quantization to discretize robot manipulation trajectories into sets of discrete sub-trajectory encodings that maximize proposed metrics on discriminative and generative informativeness.

**Strengths:**

The paper is well written and the motivations and technical details are clear.  There are detailed evaluations and the proposed method is compared to multiple SOTA approaches.  The authors also include an ablation study and highlight a comparison of of human interpretability in addition to policy performance.

**Weaknesses:**

How do the authors feel about the interpretability of manipulation concepts for human robot interaction?  Learned concepts may not be as understandable in an interaction.  I am curious about an opinion on being able to map the learned concept back to a semantic concept or how that can be integrated into the objective.  I know human intuition is covered in Table 2, and the authors state there is a weak correlation with policy performance, but there may be cases where the goal is to optimize for both.

In section 2, "partition each trajectory into semantically meaningful segments" should this be clarified on what "semantically meaningful" means?  This goes with the earlier stated motivation of moving away from human semantic discretization into self-discovered discretization that optimizes discriminative and generative informativeness.

For eq 1, how does the observed state sequence factor into the generative goal?

**Questions:**

See weaknesses

---

> ### Author Response · Authors · 2023-11-21
>
> Dear Reviewer mBdA,
>
> Thank you for your detailed review and valuable insights on our manuscript. We appreciate your recognition of the motivation of our method, your positive comments on the experimental performance, the clarity and thoroughness of the presentation, as well as the quality of evaluations.
>
> In response to your concerns, particularly regarding the interpretability of manipulation concepts for human-robot interaction and the clarification of "semantically meaningful" segments, we offer detailed explanations and additional experiments.
>
> We hope that these clarifications can help finalize your assessment and the rating of our paper. Please also let us know if you have any further questions that we need to provide additional clarifications.
>
> ***
>
> **W1: How do the authors feel about the interpretability of manipulation concepts for human robot interaction? Learned concepts may not be as understandable in an interaction. … I am curious about an opinion on being able to map the learned concept back to a semantic concept or how that can be integrated into the objective.**
>
> **A**: Thanks for the insightful comment. Before the discussion on human-robot interaction, we would like to clear the ground by a clarification of the human intuition experiment performed in the paper.
>
> Clarification: InfoCon aims at discovering concepts that can explain the “sub-goal” in manipulation processes, instead of fitting into the preference or intuition of humans. As we can see, the key states related to humans’ semantic concepts (GT key states, LLM+CLIP) do not help CoTPC policies better than key states identified by InfoCon. This indicates that InfoCon can also discover some key states which cannot be described easily using natural language, but still help us train better policies.
>
> By saying the above, we like to emphasize that the human intuition experiment is performed with the GT concepts provided by CoTPC, which only assigns a few states with key state descriptions that can be described using simple words, like, align, grasp, insert. However, it does not mean that the key states discovered by InfoCon can not be described by human language to establish interpretability.
>
> Now get back to the question “about the interpretability of manipulation concepts for human robot interaction.”
>
> We fully agree that manipulation concepts need to be meaningful enough to be assigned with robust explanations. These will help with the interpretability and transparency of the learned policies. Moreover, we want to point out that the key concepts discovered by InfoCon are understandable. In Appendix E of the updated paper, we perform an experiment to assign discovered concepts with semantically meaningful descriptions.
> For the Peg-Insertion task, the discovered concepts are:
>
> * 1. The gripper is positioned above the peg (discovered concept \#7).
> * 2. The gripper is aligned with the peg and ready to grasp (discovered concept \#5).
> * 3. The peg is grasped (discovered concept \#0).
> * 4. The peg is grasped and lifted (discovered concept \#1).
> * 5. The peg is aligned with the hole distantly (discovered concept \#4).
> * 6. The peg is aligned with the hole closely (discovered concept \#6).
> * 7. The peg is inserted half-way into the hole (discovered concept \#8).
> * 8. The peg is fully inserted (discovered concept \#2).
>
> From these experiments, we can see that the concepts are understandable. It is just that some concepts are more fine-grained and do not appear in the GT provided by CoTPC. These findings suggest that there exist fine-grained and semantically meaningful manipulation concepts that may require more words to describe but are critical for the efficiency of the manipulation policies. This also shows the effectiveness of the proposed self-supervised discovery metrics.
>
> Please also refer to Fig.8 in Appendix E of the updated manuscript for more details.
>
> Regarding “an opinion on being able to map the learned concept back to a semantic concept or how that can be integrated into the objective.”
>
> In Appendix E of the updated paper, we performed an experiment to assign discovered key concepts with semantically meaningful descriptions. Since InfoCon can automatically discover those concepts from trajectories, now we have labeled all trajectories from Peg-Insertion using the provided description. We would like to use this as a baseline to map the learned concepts back to a semantic concept in the future.
>
> Further, we would also like to prompt Large Language Models to propose semantic concepts and then optimize both the semantic compatibility generated by foundation models as well as the InfoCon criteria proposed to ensure manipulation efficiency and maximize human understanding of the discovered concepts.
>
> ***

---

> > ### Author Response · Authors · 2023-11-21
> >
> > **W2: In Section 2, "partition each trajectory into semantically meaningful segments" should this be clarified on what "semantically meaningful" means? This goes with the earlier stated motivation of moving away from human semantic discretization into self-discovered discretization that optimizes discriminative and generative informativeness.**
> >
> > **A**: Thanks for the comment. By “semantically meaningful,” we mean concepts or key states that are significant in the sense that we like to name them for better interpretability. For example, a random state showing a gripper in the air may not have much value for assigning a name.
> >
> > We also would like to point out that it does align with moving from human semantics in the sense that these “semantically meaningful” states deserve a name not because they already have a name assigned by humans, but because they are significant enough to be assigned a name, in our case the meaning is imposed by the proposed generative and discriminative goal losses.
> >
> > ***
> >
> > **W3: how does the observed state sequence factor into the generative goal?**
> >
> > **A**: Please refer to the second paragraph in Section 2.1 for a detailed description. Namely, we let the VQ-VAE partition the trajectory into a set of continuous sub-trajectories, and then the states at the boundaries of two sub-trajectories are denoted as key states. But as the training goes on, the discovered key states will also evolve to match the constraints provided by the generative and discriminative goal losses.
> >
> > The learning process involves gradually adjusting vectors within the VQVAE codebook. This means when we observe a sequence of states (sub-trajectory) under a certain label (like concept #i), we can more accurately predict its final state, based on information about #i in the VQVAE code book and the current state. This is a self-supervised process that increases the generative informativeness of vectors in the code book.

---

### Official Review · Reviewer_1PDE · 2023-11-03

**Soundness:** 3 good
**Presentation:** 2 fair
**Contribution:** 3 good
**Rating:** 8
**Confidence:** 3

**Summary:**

This manuscript proposed InfoCon, a framework that can discover concepts in manipulation tasks automatically based on information reuseness. Specifically, the authors designed an analysis system that outputting discretized concepts from an offline manipulation dataset using a few different losses: generative goal loss, discriminative goal loss. Also, the side product from the architecture is the derivative of the discriminative loss, which represents the action to perform in order to complete the task.

**Strengths:**

1. The manuscript is nicely written. I can follow most of the parts.
2. The evaluation covers a medium size of tasks and dataset, which is not perfect (not large-scale) but I believe is enough for showcasing in robot learning area.
3. The results look impressive that beat previous method marginally, achieving either the first or second best result in the whole table.

**Weaknesses:**

See question.

**Questions:**

1. I am a little bit confused about the motivation of discriminative goal loss. How is it trained and why it is useful for extracting information from the input states?
2. I am not fully confident about the fairness of the comparison. It seems to me that methods including LLM+CLIP are zero-shot learning process that does not require the training data to be seen. Am I correct or it's actually not the case? If so, I would suggest to separating them in the comparison. However, I still think the proposed method has technical contributions that is worth to present.
3. Is there any real-world example to indicate the effectiveness?

---

> ### Author Response · Authors · 2023-11-21
>
> Dear Reviewer 1PDE,
>
> Thank you for your comprehensive review and the positive remarks on our manuscript. We highly appreciate your recognition of our efforts in proposing an automatic concept discovery framework and analysis system along with it.
>
> In response to your insightful questions, we have addressed each of your concerns in detail in the following. We hope that these clarifications can help finalize your assessment and the rating of our paper. Please also let us know if you have any further questions that we need to provide additional clarifications.
>
> ***
>
> **Q1.1: I am a little bit confused about the motivation of discriminative goal loss.**
>
> **A**: The discriminative goal is motivated by the observation that given a (physical) state (e.g., a hand holding a cup), one can assign a compatibility score to this state with varying goals. For example, if the goal is “getting water to drink,” then the compatibility score of the state with the goal would be high; otherwise, if the goal is “performing yoga,” the score would be low. In other words, a concept as a discriminative goal shall be able to help distinguish the state that is within the process of achieving this goal or not.
>
> The above motivates us to formulate the concept as a discriminative goal, in the sense that the concept can be transformed into a discriminator conditioned on the concept.
>
> Moreover, still with the example of a state “a hand holding a cup” and another “a hand holding a cup placed under the faucet,” the discriminator instantiated by the concept “getting water to drink” shall assign a higher score of the latter than the former one. This phenomenon illustrates that a concept in its discriminative form shall inform the change in the state (action) via its change in the score (gradient). This gives us the second term regarding the discriminative goal.
>
> ***
>
> **Q1.2: How is it trained and why it is useful for extracting information from the input states?**
>
> **A**: How it is trained:
>
> Each manipulation concept is related to a vector. We decode this vector into a compatibility (scalar) function representing the discriminative goal of this concept via a hyper-network, which takes in the vector and outputs the weights of the concept-conditioned discriminator.
>
> The concept-conditioned discriminator is then trained with two loss terms regarding the concept as a discriminative goal. The concept-conditioned discriminator function shall satisfy the two aspects described in Q1.1: grading states and providing states transition information using its gradient field. In other words, it should assign high scores to the states that are in the process of achieving this goal, and enable good prediction of the action with its gradient. These two losses are instantiated in the form of cross-entropy and L-2 reconstruction.
>
> Why it is useful:
>
> With the above loss terms, the VQVAE structure employed in our pipeline will get training signals to adjust both its encoder and the codebooks and learn to discover meaningful concepts (in the form of codes in the codebook).
>
> The learning dynamic consists of two aspects. First, the encoder and the codebook will adjust to assign a state to a discriminator that assigns the highest score to it, which encourages clustering according to the discriminator. On the other hand, the discriminator shall encourage meaningful clusterings, otherwise, its gradients will not be informative of the action. For example, if the discriminator assigns all states to a single concept, then it is uniformly 1 over the state space, and its gradient is minimally informative of the action (state change). These correlations ensure that the whole training process is stable and the concept learned is non-trivial, promoting the usefulness of the learned concepts.
>
> ***

---

> > ### Author Response · Authors · 2023-11-21
> >
> > **Q2: It seems to me that methods including LLM+CLIP are zero-shot learning process that does not require the training data to be seen. … If so, I would suggest to separating them in the comparison. However, I still think the proposed method has technical contributions that is worth to present.**
> >
> > **A**: We appreciate your insight regarding “zero-shot learning” and understand that it suggests methods like LLM+CLIP are trained with datasets distinct from the demonstrations we utilize. We have revised the paper to separate them in the comparison to make this point more explicit.
> >
> > Our goal in putting these methods here is to examine how large foundation models, including LLMs and VLMs, can help with manipulation concept discovery. Our experiment primarily aims to contrast methods that are rooted in human concepts against those that are intrinsically linked to the inherent properties of the data. In this context, LLM+CLIP and ground truth (GT) key states are representatives of approaches grounded in human concepts, while AWE and InfoCon exemplify methods that are fundamentally based on data itself. This selection of baselines is to see the differences in concepts derived from the data itself with concepts aligned with human semantics.
> >
> > ***
> >
> > **Q3: Is there any real-world example to indicate the effectiveness?**
> >
> > **A**: Thanks for the comment. Due to the time limit, we have run the proposed InfoCon on human pose sequences estimated from real-world videos to check its efficiency in discovering human motion concepts. A qualitative result is presented in Fig. 9. With this experiment, we show the potential of the proposed InfoCon in discovering key states from real-world data, but we are interested in training InfoCon on large-scale data when time permits.

---

### Author Response · Authors · 2023-11-21
**Updated Manuscript and Response to All Reviewers**

We sincerely thank all reviewers for their insightful feedback and constructive comments on our manuscript. We are grateful to note that the reviewers appreciate the novelty of our approach in addressing the manipulation concept discovery problem (R#1, R#2, R#3, R#4). Moreover, it is recognized that our work offers a fresh perspective and a substantial solution to the growing field of robot learning (R#1, R#2, R#3, R#4). The reviewers have also echoed our solid experiment results across a range of comprehensive tasks (R#1, R#2, R#3). Additionally, we appreciate the acknowledgment of our paper's presentation and writing quality (R#1, R#2).

Also, based on these comments, we provide more experiments and update the manuscript. All revisions are highlighted in blue color in the new manuscript. The modifications are summarized as follows:
1. **Section 2 Problem Setup**: Add an explanation of “semantically meaningful partition” (R#2) and details of evaluating manipulation concepts on physical manipulation benchmarks (R#3).
2. **Section 3.2, Table 1**: Clarify the baselines for manipulation concept discovery with zero-shot and non-zero-shot methods (R#1).
3. **Appendix A.5**: Extra experiments to verify the effectiveness of reconstruction pretraining (R#3).
4. **Appendix E** Extra experiments on alignment with human concepts. (R#2)
5. **Appendix F** Extra experiments and visualization of applying InfoCon with real-world data. (R#1, R#3)

Our sincere thanks go out once again to all reviewers for their valuable contributions towards enhancing our manuscript. We have addressed the queries and points raised by each reviewer. Should there be a need for further clarification to assist in advancing our score, please do not hesitate to inform us.

Thank you for your review!

---

### Meta-Review · Area_Chair_9Tpk · 2023-12-10

**Metareview:**

This paper presents an unsupervised learning method for keyframe detection and segmentation of trajectories into a sequence of sub-tasks/concepts. The method relies on three losses/requirements:  (1) that for any given state, the task is predictive of the next keyframe, (2) for every task, there is a compatibility function that describes how compatible a given state is with the task, and (3) for every task, the gradient of the compatibility function of that task is predictive of the next action. Of these losses, I find (1) and (2) have appeared in similar forms in prior literature on task segmentation. The paper evaluates the quality of detected keyframes according to how well a downstream policy that has access to these keyframes performs in terms of task completion and success rate, where the proposed method does well compared to baselines. It also compares the discovered keyframes with ones labeled by humans, where high correlation is shown.

While this is an interesting paper, I'm afraid its related works section misses a large number of relevant papers and ideas on unsupervised skill discovery and changepoint detection. For example:

[1] Adaptive Skip Intervals: Temporal Abstraction for Recurrent Dynamical Models, Nietz et al.
[2] Time-Agnostic Prediction: Predicting Predictable Video Frames, Jayaraman et al
[3] Bottom-Up Skill Discovery From Unsegmented Demonstrations for Long-Horizon Robot Manipulation, Zhu et al
[4] Adaptive Gaussian Process Change Point Detection, Caldarelli et al
[5] Keyframing the Future: Keyframe Discovery for Visual Prediction and Planning, Pertsch et al

And that doesn't include relevant literature cited in these papers.

**Justification For Why Not Higher Score:**

I don't think the paper has done a good job in terms of citing related work, so I cannot recommend it as a spotlight paper.

**Justification For Why Not Lower Score:**

I don't think it merits rejection.

---

### Decision · Program_Chairs · 2024-01-16

Accept (poster)